# 🔥 PROMETHEUS: INDUCING FINE-GRAINED EVALUATION CAPABILITY IN LANGUAGE MODELS

**Seungone Kim**[1,2*†]  **Jamin Shin**[2,3*†]  **Yejin Cho**[1*†]  **Joel Jang**[4]  **Shayne Longpre**[5]
**Hwaran Lee**[2,3]  **Sangdoo Yun**[2,3]  **Seongjin Shin**[3]  **Sungdong Kim**[1,2,3]
**James Thorne**[1]  **Minjoon Seo**[1†]

[1]KAIST AI   [2]NAVER AI Lab   [3]NAVER Cloud   [4]University of Washington   [5]MIT

{seungone, yejin_cho, minjoon}@kaist.ac.kr   jamin.shin@outlook.com

## ABSTRACT

Recently, using a powerful proprietary Large Language Model (LLM) (e.g., GPT-4) as an evaluator for long-form responses has become the de facto standard. However, for practitioners with large-scale evaluation tasks and custom criteria in consideration (e.g., child-readability), using proprietary LLMs as an evaluator is unreliable due to the closed-source nature, uncontrolled versioning, and prohibitive costs. In this work, we propose PROMETHEUS, a fully open-source LLM that is on par with GPT-4's evaluation capabilities when the appropriate reference materials (reference answer, score rubric) are accompanied. We first construct the FEEDBACK COLLECTION, a new dataset that consists of 1K fine-grained score rubrics, 20K instructions, and 100K responses and language feedback generated by GPT-4. Using the FEEDBACK COLLECTION, we train PROMETHEUS, a 13B evaluator LLM that can assess any given long-form text based on *customized* score rubric provided by the user. Experimental results show that PROMETHEUS scores a Pearson correlation of 0.897 with human evaluators when evaluating with 45 customized score rubrics, which is on par with GPT-4 (0.882), and greatly outperforms ChatGPT (0.392). Furthermore, measuring correlation with GPT-4 with 1222 customized score rubrics across four benchmarks (MT Bench, Vicuna Bench, Feedback Bench, Flask Eval) shows similar trends, bolstering PROMETHEUS's capability as an evaluator LLM. Lastly, PROMETHEUS achieves the highest accuracy on two human preference benchmarks (HHH Alignment & MT Bench Human Judgment) compared to open-sourced reward models explicitly trained on human preference datasets, highlighting its potential as an universal reward model. We will open-source our code, dataset, and model [1].

## 1 INTRODUCTION

Evaluating the quality of machine-generated text has been a long-standing challenge in Natural Language Processing (NLP) and remains especially essential in the era of Large Language Models (LLMs) to understand their properties and behaviors (Liang et al., 2022; Chang et al., 2023; Zhong et al., 2023; Chia et al., 2023; Holtzman et al., 2023). Human evaluation has consistently been the predominant method, for its inherent reliability and capacity to assess nuanced and subjective dimensions in texts. In many situations, humans can naturally discern the most important factors of assessment, such as brevity, creativity, tone, and cultural sensitivities. On the other hand, conventional automated evaluation metrics (e.g., BLEU, ROUGE) cannot capture the depth and granularity of human evaluation (Papineni et al., 2002; Lin, 2004b; Zhang et al., 2019; Krishna et al., 2021).

Applying LLMs (e.g. GPT-4) as an evaluator has received substantial attention due to its potential parity with human evaluation (Chiang & yi Lee, 2023; Dubois et al., 2023; Li et al., 2023; Liu et al.,

---

*denotes equal contribution. Work was done while Seungone was interning at NAVER AI Lab.

†Corresponding authors

[1]https://kaistai.github.io/prometheus/

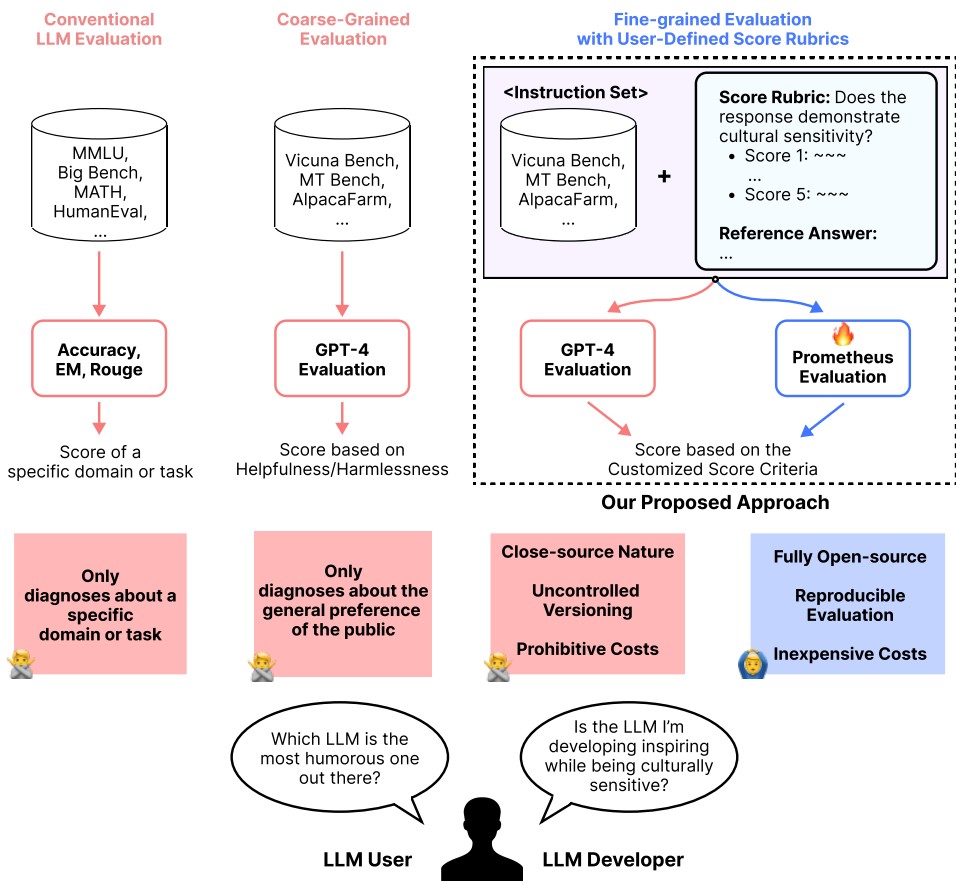

Figure 1: Compared to conventional, coarse-grained LLM evaluation, we propose a fine-grained approach that takes user-defined score rubrics as input.

2023; Peng et al., 2023; Zheng et al., 2023; Ye et al., 2023b; Min et al., 2023). Initial investigations and observations indicate that, when aptly prompted, LLMs can emulate the fineness of human evaluations. However, while the merits of using proprietary LLMs as an evaluation tool are evident, there exist some critical disadvantages:

1. **Closed-source Nature**: The proprietary nature of LLMs brings transparency concerns as internal workings are not disclosed to the broader academic community. Such a lack of transparency hinders collective academic efforts to refine or enhance its evaluation capabilities. Furthermore, this places fair evaluation, a core tenet in academia, under control of for-profit entity and raises concerns about neutrality and autonomy.

2. **Uncontrolled Versioning**: Proprietary models undergo version updates that are often beyond the users' purview or control (Pozzobon et al., 2023). This introduces a reproducibility challenge. As reproducibility is a cornerstone of scientific inquiry, any inconsistency stemming from version changes can undermine the robustness of research findings that depend on specific versions of the model, especially in the context of evaluation.

3. **Prohibitive Costs**: Financial constraints associated with LLM APIs are not trivial. For example, evaluating four LLMs variants across four sizes (ranging from 7B to 65B) using GPT-4 on 1000 evaluation instances can cost over $2000. Such scaling costs can be prohibitive, especially for academic institutions or researchers operating on limited budgets.

Despite these limitations, proprietary LLMs such as GPT-4 are able to evaluate scores based on *customized* score rubrics. Specifically, current resources are confined to *generic*, single-dimensional evaluation metrics that are either too domain/task-specific (e.g. EM, Rouge) or coarse-grained (e.g. helpfulness/harmlessness (Dubois et al., 2023; Chiang et al., 2023; Liu et al., 2023) as shown in left-

side of Figure 1. For instance, AlpacaFarm's (Dubois et al., 2023) prompt gives a single definition of preference, asking the model to choose the model response that is *generally* preferred. However, response preferences are subject to variation based on specific applications and values. In real-world scenarios, users may be interested in *customized* rubric such as "Which LLM generates responses that are playful and humorous" or "Which LLM answers with particularly care for cultural sensitivities?" Yet, in our initial experiments, we observe that even the largest open-source LLM (70B) is insufficient to evaluate based on a customized score rubric compared to proprietary LLMs.

To this end, we propose PROMETHEUS, a 13B LM that aims to induce fine-grained evaluation capability of GPT-4, while being open-source, reproducible, and inexpensive. We first create the FEEDBACK COLLECTION, a new dataset that is crafted to encapsulate diverse and fine-grained user assessment score rubric that represent realistic user demands (example shown in Figure 2). We design the FEEDBACK COLLECTION with the aforementioned consideration in mind, encompassing thousands of unique preference criteria encoded by a user-injected score rubric. Unlike prior feedback datasets (Ye et al., 2023a; Wang et al., 2023a), it uses *custom*, not *generic* preference score rubric, to train models to flexibly generalize to practical and diverse evaluation preferences. Also, to best of our knowledge, we are first to explore the importance of including various reference materials – particularly the 'Reference Answers' – to effectively induce fine-grained evaluation capability.

We use the FEEDBACK COLLECTION to fine-tune Llama-2-Chat-13B in creating PROMETHEUS. On 45 customized score rubrics sampled across three test sets (MT Bench, Vicuna Bench, Feedback Bench), PROMETHEUS obtains a Pearson correlation of 0.897 with human evaluators, which is similar with GPT-4 (0.882), and has a significant gap with GPT-3.5-Turbo (0.392). Unexpectely, when asking human evaluators to choose a feedback with better quality in a pairwise setting, PROMETHEUS was preferred over GPT-4 in 58.67% of the time, while greatly outperformed GPT-3.5-Turbo with a 79.57% win rate. Also, when measuring the Pearson correlation with GPT-4 evaluation across 1222 customized score rubrics across 4 test sets (MT Bench, Vicuna Bench, Feedback Bench, Flask Eval), PROMETHEUS showed higher correlation compared to GPT-3.5-Turbo and Llama-2-Chat 70B. Lastly, when testing on 2 unseen human preference datasets (MT Bench Human Judgments, HHH Alignment), PROMETHEUS outperforms two state-of-the-art reward models and GPT-3.5-Turbo, highlighting its potential as an universal reward model.

## 2 RELATED WORK

**Reference-based text evaluation** Previously, model-free scores that evaluate machine-generated text based on a golden candidate reference such as BLEU (Papineni et al., 2002) and ROUGE (Lin, 2004a) scores were used as the dominant approach. In recent years, model-based approaches have been widely adopted such as BERTScore (Zhang et al., 2019), BLEURT (Sellam et al., 2020), and BARTScore (Yuan et al., 2021) which are able to capture *semantic* information rather than only evaluating on *lexical* components. Recently, Krishna et al. (2021) reported limitations in reference-based metrics, such as ROUGE, observing that they are not reliable for evaluation.

**LLM-based text evaluation** Recent work has used GPT-4 or a fine-tuned critique LLM as an evaluator along a single dimension of "preference" (Chiang & yi Lee, 2023; Li et al., 2023; Dubois et al., 2023; Zheng et al., 2023; Liu et al., 2023). For instance, AlpacaFarm (Dubois et al., 2023) asks the model to select "which response is better based on your judgment and based on your own preference" Another example is recent work that showed ChatGPT can outperform crowd-workers for text-annotation tasks (Gilardi et al., 2023; Chiang & yi Lee, 2023). Wang et al. (2023b) introduced PandaLM, a fine-tuned LLM to evaluate the generated text and explain its reliability on various preference datasets. Similarly, Ye et al. (2023a) and Wang et al. (2023a) create critique LLMs. However, the correct preference is often subjective and depends on applications, cultures, and objectives, where degrees of brevity, formality, honesty, creativity, and political tone, among many other potentially desirable traits that may vary (Chiang & yi Lee, 2023). While GPT-4 is unreliable due to its close-source nature, uncontrolled versioning, and prohibitive costs, it was the *only* option explored for fine-grained and customized evaluation based on the score rubric (Ye et al., 2023b). On the contrary, we train, to best of our knowledge, the first evaluator sensitive to thousands of unique preference criteria, and show it significantly outperforms uni-dimensional preference evaluators in a number of realistic settings. Most importantly, compared to previous work, we strongly argue

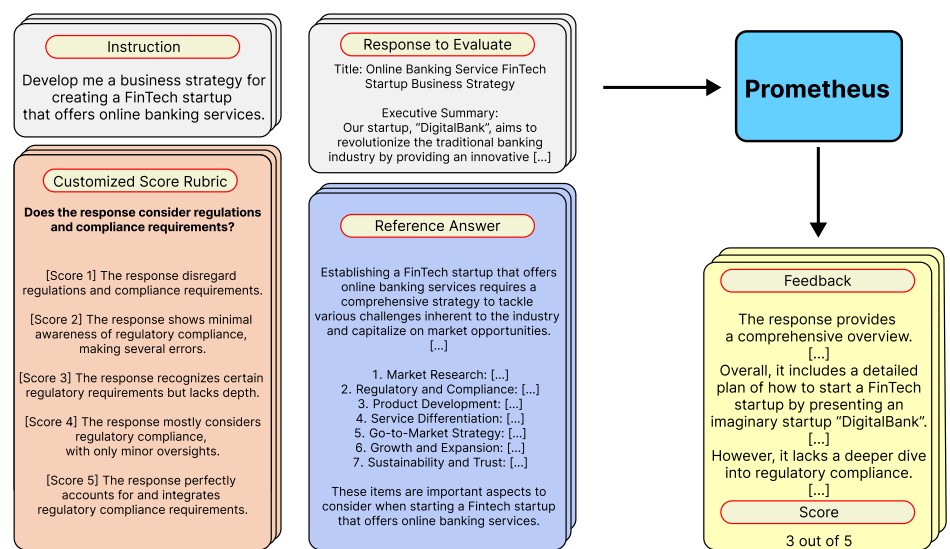

Figure 2: The individual components of the FEEDBACK COLLECTION. By adding the appropriate reference materials (Score Rubric and Reference Answer) and training on GPT-4's feedback, we show that we could obtain a strong open-source evaluator LM.

Table 1: Information about our training dataset FEEDBACK COLLECTION. Note that there are 20 instructions accompanied for each score rubric, leading to a total number of 20K. Also, there is a score 1-5 response and feedback for each instruction, leading to a total number of 100K.

| Evaluation Mode | Data | # Score Rubrics | # Instructions & Reference Answer | # Responses & Feedback |
|---|---|---|---|---|
| **Absolute Evaluation** | FEEDBACK COLLECTION | 1K (Fine-grained & Customized) | Total 20K (20 for each score rubric) | Total 100K (5 for each instruction; 20K for each score within 1-5) |

the importance of appending reference materials (score rubric and reference answer) in addition to fine-tuning on the feedback in order to effectively induce fine-grained evaluation capability.

## 3 THE FEEDBACK COLLECTION DATASET

While previous work has demonstrated that fine-tuning on feedback is effective for improving LMs to function as a critique (Ye et al., 2023a; Wang et al., 2023a), the datasets used in previous work are not directly applicable for improving LMs to function as a fine-grained *evaluator*. We thus introduce the FEEDBACK COLLECTION, a new dataset for the sole purpose of fine-tuning an open-sourced evaluator LLM. Our 4 main considerations during dataset construction are: (1) including as many reference materials (i.e. reference answer, and scoring rubric) as possible, (2) maintaining a uniform length among the reference answers for each score (1 to 5) to prevent undesired length bias, (3) maintaining a uniform score distribution to prevent undesired decision bias, and (4) limiting the scope of our instructions and responses to realistic situations where a user is interacting with a LLM.

Taking these into consideration, we construct each instance within the FEEDBACK COLLECTION to encompass four components for the *input* (instruction, response to evaluate, customized score rubric, reference answer) and two components in the *output* (feedback, score). An example of an instance is shown in Figure 2 and the number of each component is shown in Table 1.

The four components for the input are as follows:

1. **Instruction**: An instruction that a user would prompt to an arbitrary LLM.

2. **Response to Evaluate**: A response to the instruction that the evaluator LM has to evaluate.

3. **Customized Score Rubric**: A specification of novel criteria decided by the user. The evaluator should focus on this aspect during evaluation. The rubric consists of (1) a description of the criteria and (2) a description of each scoring decision (1 to 5).

4. **Reference Answer**: A reference answer that would receive a score of 5. Instead of requiring the evaluator LM to solve the instruction, it enables the evaluator to use the mutual information between the reference answer and the response to make a scoring decision.

The two components for the output are as follows:

1. **Feedback**: A rationale of why the provided response would receive a particular score. This is analogous to Chain-of-Thoughts (CoT), making the evaluation process interpretable.

2. **Score**: An integer score for the provided response that ranges from 1 to 5.

Each instance has an accompanying scoring rubric and reference answer upon the instruction in order to include as much reference material as possible. Also, we include an equal number of 20K instances for each score in the range of 1 to 5, preventing undesired decision bias while training the evaluator LLM. A detailed analysis of the FEEDBACK COLLECTION dataset is in Appendix D.

## 3.1 DATASET CONSTRUCTION PROCESS

We construct a large-scale FEEDBACK COLLECTION dataset by prompting GPT-4. Specifically, the collection process consists of (1) the curation of 50 initial seed rubrics, (2) the expansion of 1K new score rubrics through GPT-4, (3) the augmentation of realistic instructions, and (4) the augmentation of the remaining components in the training instances (i.e. responses including the reference answers, feedback, and scores). Figure 6 shows the overall augmentation process.

**Step 1: Creation of the Seed Rubrics** We begin with the creation of a foundational seed dataset of scoring rubrics. Each author curates a detailed and fine-grained scoring rubric that each personnel considers pivotal in evaluating outputs from LLMs. This results in an initial batch of 50 seed rubrics.

**Step 2: Augmenting the Seed Rubrics with GPT-4** Using GPT-4 and our initial seed rubrics, we expand the score rubrics from the initial 50 to a more robust and diverse set of 1000 score rubrics. Specifically, by sampling 4 random score rubrics from the initial seed, we use them as demonstrations for in-context learning (ICL), and prompt GPT-4 to brainstorm a new novel score rubric. Also, we prompt GPT-4 to paraphrase the newly generated rubrics in order to ensure PROMETHEUS could generalize to the similar score rubric that uses different words. We iterate the brainstorming → paraphrasing process for 10 rounds. The detailed prompt used for this procedure is in Appendix J.

**Step 3: Crafting Novel Instructions related to the Score Rubrics** With a comprehensive dataset of 1000 rubrics at our disposal, the subsequent challenge was to craft pertinent training instances. For example, a score rubric asking "Is it formal enough to send to my boss" is not related to a math problem. Considering the need for a set of instructions closely related to the score rubrics, we prompt GPT-4 to generate 20K unique instructions that are highly relevant to the given score rubric.

**Step 4: Crafting Training Instances** Lastly, we sequentially generate a response to evaluate and corresponding feedback by prompting GPT-4 to generate each component that will get a score of $i$ ($1 \leq i \leq 5$). This leads to 20 instructions for each score rubric, and 5 responses & feedback for each instruction. To eliminate the effect of decision bias when fine-tuning our evaluator LM, we generate an equal number of 20K responses for each score. Note that for the response with a score of 5, we generate two distinctive responses so we could use one of them as an input (reference answer).

## 3.2 FINE-TUNING AN EVALUATOR LM

Using the FEEDBACK COLLECTION dataset, we fine-tune Llama-2-Chat (7B & 13B) and obtain PROMETHEUS to induce fine-grained evaluation capability. Similar to Chain-of-Thought Fine-tuning (Ho et al., 2022; Kim et al., 2023a), we fine-tune to sequentially generate the feedback and then the score. We highlight that it is important to include a phrase such as '[RESULT]' in between the feedback and the score to prevent degeneration during inference. We include the details of fine-tuning and inference in Appendix F.

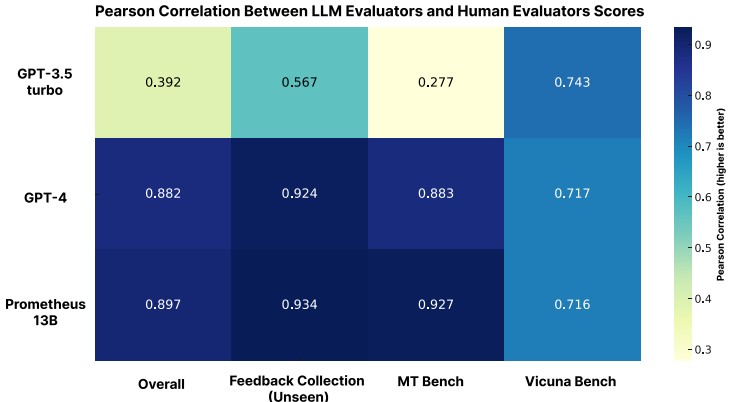

Figure 3: The Pearson correlation between scores from human annotators and the score from GPT-3.5-Turbo, Prometheus, and GPT-4 on 45 customized score rubrics from the Feedback Bench, Vicuna Bench, and MT Bench. PROMETHEUS shows a high correlation with human evaluators.

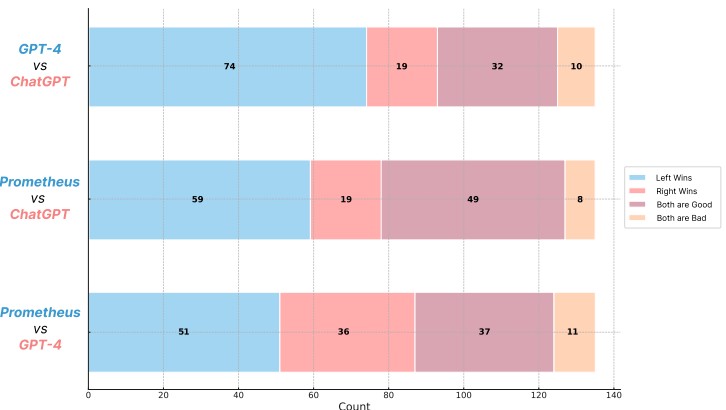

Figure 4: Pairwise comparison of the quality of the feedback generated by GPT-4, PROMETHEUS and GPT-3.5-Turbo. Annotators are asked to choose which feedback is better at assessing the given response. PROMETHEUS shows a win-rate of 58.62% over GPT-4 and 79.57% over GPT-3.5-Turbo.

## 4 EXPERIMENTAL SETTING: EVALUATING AN EVALUATOR LM

In this section, we explain our experiment setting, including the list of experiments, metrics, and baselines that we use to evaluate *fine-grained* evaluation capabilities of an evaluator LLM. Compared to measuring the instruction-following capability of a LLM, it is not straightforward to directly measure the capability to evaluate. Therefore, we use human evaluation and GPT-4 evaluation as a standard and measure how similarly our evaluator model and baselines could closely simulate them. We mainly employ two types of evaluation methods: *Absolute Grading* and *Ranking Grading*.

Our experimental setting including "List of Experiments", "Metrics" and "Baselines" are in Appendix B.

## 5 EXPERIMENTAL RESULTS

### 5.1 CAN PROMETHEUS CLOSELY SIMULATE HUMAN EVALUATION?

**Correlation with Human Scoring** We first compare the correlation between human annotators and our baselines using 45 instances each with an unique customized score rubric, namely the FEEDBACK BENCH (Unseen Score Rubric subset), MT Bench (Zheng et al., 2023), and Vicuna Bench (Chiang et al., 2023). The results are shown in Figure 3, showing that PROMETHEUS is

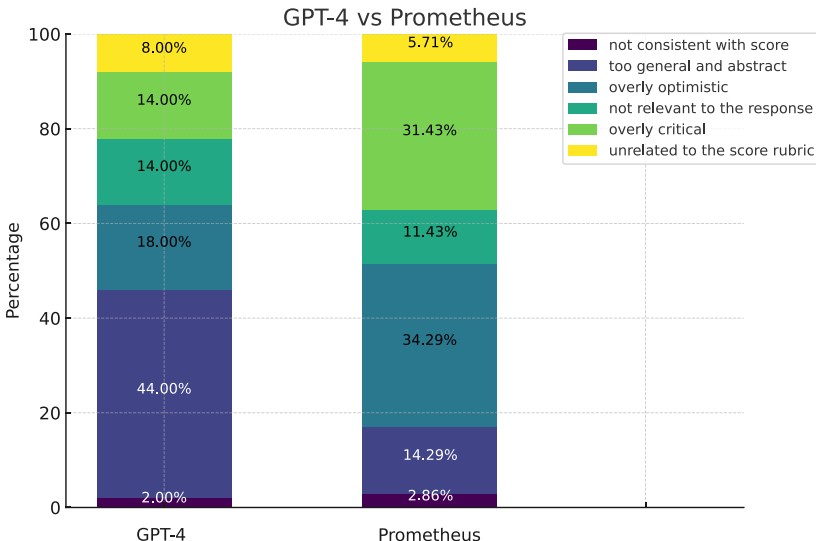

Figure 5: The reason why GPT-4's or Prometheus's feedback was not chosen over the other. PROMETHEUS generates less abstract and general feedback, but tends to write overly critical ones.

on par with GPT-4 across all the three evaluation datasets, where PROMETHEUS obtains a 0.897 Pearson correlation, GPT-4 obtains 0.882, and GPT-3.5-Turbo obtains 0.392.

**Pairwise Comparison of the Feedback with Human Evaluation**   To validate the effect of whether PROMETHEUS generates helpful/meaningful feedback in addition to its scoring decision, we ask human annotators to choose a better feedback. The results are shown in Figure 4, showing that PROMETHEUS is preferred over GPT-4 58.62% of the times, and over GPT-3.5-Turbo 79.57% of the times. This shows that PROMETHEUS's feedback is also meaningful and helpful.

**Analysis of Why Prometheus's Feedback was Preferred**   In addition to a pairwise comparison of the feedback quality, we also conduct an analysis asking human annotators to choose why they preferred one feedback over the other by choosing at least one of the comprehensive 6 options ("rejected feedback is not consistent with its score" / "too general and abstract" / "overly optimistic" / "not relevant to the response" / "overly critical" / "unrelated to the score rubric"). In Figure 5, we show the percentage of why each evaluator LLM (GPT-4 and PROMETHEUS) was rejected. It shows that while GPT-4 was mainly not chosen due to providing general or abstract feedback, PROMETHEUS was mainly not chosen because it was too critical about the given response. Based on this result, we conclude that whereas GPT-4 tends to be more neutral and abstract, PROMETHEUS shows a clear trend of expressing its opinion of whether the given response is good or not. We conjecture this is an effect of directly fine-tuning PROMETHEUS to ONLY perform fine-grained evaluation, essentially converting it to an evaluator rather than a generator. We include (1) additional results of analyzing "PROMETHEUS *vs* GPT-3.5-Turbo" and "GPT-4 *vs* GPT-3.5-Turbo" in Appendix H and (2) a detailed explanation of the experimental setting of human evaluation in Appendix M.

### 5.2   CAN PROMETHEUS CLOSELY SIMULATE GPT-4 EVALUATION?

**Correlation with GPT-4 Scoring**   We compare the correlation between GPT-4 evaluation and our baselines using 1222 score rubrics across 2360 instances from the FEEDBACK BENCH (Seen and Unseen Score Rubric Subset), Vicuna Bench (Chiang et al., 2023), MT Bench (Zheng et al., 2023), and Flask Eval (Ye et al., 2023b) in an absolute grading scheme. Note that for the FEEDBACK BENCH, we measure the correlation with the scores **augmented** from GPT-4-0613, and for the other 3 datasets, we measure the correlation with the scores acquired by **inferencing** GPT-4-0613.

The results on these benchmarks are shown across Table 2 and Table 3.

Table 2: Pearson, Kendall-Tau, Spearman correlation with data generated by GPT-4-0613. All scores were sampled across 3 inferences. The best comparable statistics are **bolded** and second best underlined.

| Evaluator LM | FEEDBACK COLLECTION TEST SET (GENERATED BY GPT-4-0613) | | | | | |
| | SEEN CUSTOMIZED RUBRICS | | | UNSEEN CUSTOMIZED RUBRIC | | |
| | Pearson | Kendall-Tau | Spearman | Pearson | Kendall-Tau | Spearman |
|---|---|---|---|---|---|---|
| LLAMA2-CHAT 7B | 0.485 | 0.422 | 0.478 | 0.463 | 0.402 | 0.465 |
| LLAMA2-CHAT 13B | 0.441 | 0.387 | 0.452 | 0.450 | 0.379 | 0.431 |
| LLAMA2-CHAT 70B | 0.572 | 0.491 | 0.564 | 0.558 | 0.477 | 0.549 |
| LLAMA2-CHAT 13B + COARSE. | 0.482 | 0.406 | 0.475 | 0.454 | 0.361 | 0.427 |
| PROMETHEUS 7B | 0.860 | **0.781** | **0.863** | 0.847 | 0.767 | 0.849 |
| PROMETHEUS 13B | **0.861** | 0.776 | 0.858 | **0.860** | **0.771** | **0.858** |
| GPT-3.5-TURBO-0613 | 0.636 | 0.536 | 0.617 | 0.563 | 0.453 | 0.521 |
| GPT-4-0314 | 0.754 | 0.671 | 0.762 | 0.753 | 0.673 | 0.761 |
| GPT-4-0613 | 0.742 | 0.659 | 0.747 | 0.743 | 0.660 | 0.747 |
| GPT-4 (RECENT) | 0.745 | 0.659 | 0.748 | 0.733 | 0.641 | 0.728 |

Table 3: Pearson, Kendall-Tau, Spearman correlation with scores sampled from GPT-4-0613 across 3 inferences. Note that GPT-4-0613 was sampled 6 times in total to measure self-consistency. The best comparable statistics are **bolded** and second best underlined among baselines. We include GPT-4 as reference to show it self-consistency when inferenced multiple times.

| Evaluator LM | VICUNA BENCH | | | MT BENCH | | | FLASK EVAL | | |
| | Pearson | Kendall-Tau | Spearman | Pearson | Kendall-Tau | Spearman | Pearson | Kendall-Tau | Spearman |
|---|---|---|---|---|---|---|---|---|---|
| LLAMA2-CHAT 7B | 0.175 | 0.143 | 0.176 | 0.132 | 0.113 | 0.143 | 0.271 | 0.180 | 0.235 |
| LLAMA2-CHAT 13B | 0.211 | 0.203 | 0.253 | -0.020 | -0.029 | -0.038 | 0.265 | 0.182 | 0.235 |
| LLAMA2-CHAT 70B | 0.376 | 0.318 | 0.391 | 0.226 | 0.175 | 0.224 | 0.336 | 0.267 | 0.346 |
| LLAMA2-CHAT 13B + COARSE. | 0.307 | 0.196 | 0.245 | 0.417 | 0.328 | 0.420 | **0.517** | **0.349** | 0.451 |
| PROMETHEUS-7B | 0.457 | **0.365** | **0.457** | 0.293 | 0.216 | 0.295 | 0.367 | 0.285 | 0.371 |
| PROMETHEUS-13B | **0.466** | 0.346 | 0.429 | **0.473** | **0.341** | **0.451** | 0.467 | 0.345 | **0.455** |
| GPT-3.5-TURBO-0613 | 0.270 | 0.187 | 0.232 | 0.275 | 0.202 | 0.267 | 0.422 | 0.299 | 0.371 |
| GPT-4-0314 | 0.833 | 0.679 | 0.775 | 0.857 | 0.713 | 0.849 | 0.785 | 0.621 | 0.747 |
| GPT-4-0613 | 0.925 | 0.783 | 0.864 | 0.952 | 0.834 | 0.927 | 0.835 | 0.672 | 0.798 |
| GPT-4 (RECENT) | 0.932 | 0.801 | 0.877 | 0.944 | 0.812 | 0.914 | 0.832 | 0.667 | 0.794 |

In Table 2, the performance of LLAMA-2-CHAT 13B degrades over the 7B model and slightly improves when scaled up to 70B size, indicating that naively increasing the size of a model does not necessarily improve an LLM's evaluation capabilities. On the other hand, PROMETHEUS 13B shows a +0.420 and +0.397 improvement over its base model LLAMA2-CHAT 13B in terms of Pearson correlation on the seen and unseen rubric set, respectively. Moreover, it even outperforms LLAMA2-CHAT 70B, GPT-3.5-TURBO-0613, and different versions of GPT-4. We conjecture the high performance of PROMETHEUS is mainly because the instructions and responses within the test set might share a similar distribution with the train set we used (simulating a scenario where a user is interacting with a LLM) even if the score rubric holds unseen. Also, training on feedback derived from coarse-grained score rubrics (denoted as LLAMA2-CHAT 13B + COARSE) only slightly improves performance, indicating the importance of training on a wide range of score rubric is important to handle customized rubrics that different LLM user or researcher would desire.

In Table 3, the trends of LLAMA2-CHAT among different sizes hold similar; simply increasing size does not greatly improve the LLM's evaluation capabilities. On these benchmarks, PROMETHEUS shows a +0.255, +0.493, and +0.202 improvement over its base model LLAMA2-CHAT-13B in terms of Pearson correlation on the Vicuna Bench, MT Bench, and Flask Eval dataset, respectively. While PROMETHEUS outperforms LLAMA2-CHAT 70B and GPT-3.5-TURBO-0613, it lacks behind GPT-4. We conjecture that this might be because the instructions from the FEEDBACK COLLECTION and these evaluation datasets have different characteristics; the FEEDBACK COLLECTION are relatively long and detailed (e.g., I'm a city planner ... I'm looking for a novel and progressive

Table 4: Human Agreement accuracy among ranking datasets. The best comparable statistics are **bolded**.

| Evaluator LM | HHH Alignment | | | | | MT Bench Human Judg. |
| --- | --- | --- | --- | --- | --- | --- |
| | Help. | Harm. | Hon. | Other | Total Avg. | Human Preference |
| Random | 50.00 | 50.00 | 50.00 | 50.00 | 50.00 | 34.26 |
| StanfordNLP Reward Model | 69.49 | 60.34 | 52.46 | 51.16 | 58.82 | 44.79 |
| ALMOST Reward Model | 74.58 | 67.24 | **78.69** | 86.05 | 76.02 | 49.90 |
| Llama2-Chat 7B | 66.10 | 81.03 | 70.49 | 74.42 | 72.85 | 51.78 |
| Llama2-Chat 13B | 74.58 | 87.93 | 55.74 | 79.07 | 73.76 | 52.34 |
| Llama2-Chat 70B | 66.10 | **89.66** | 67.21 | 74.42 | 74.21 | 53.67 |
| Llama2-Chat 13B + Coarse. | 68.74 | 68.97 | 65.57 | 67.44 | 67.42 | 46.89 |
| GPT-3.5-Turbo-0613 | 76.27 | 87.93 | 67.21 | 86.05 | 78.73 | 57.12 |
| Prometheus 7B | 69.49 | 84.48 | **78.69** | **90.70** | **80.09** | 55.14 |
| Prometheus 13B | **81.36** | 82.76 | 75.41 | 76.74 | 79.19 | **57.72** |
| GPT-4-0613 | 91.53 | 93.10 | 85.25 | 83.72 | 88.69 | 63.87 |

solution to handle traffic congestion and air problems derived from population increase), while the datasets used for evaluation hold short (e.g., Can you explain about quantum mechanics?).

On the other hand, it is important to note that on the Flask Eval dataset, Llama2-Chat 13B + Coarse (specifically trained with the Flask Eval dataset) outperforms Prometheus. This indicates that training directly on the evaluation dataset might be the best option to acquire a task-specific evaluator LLM, and we further discuss this in Section C.5.

## 6 Can Prometheus Function as a Reward Model?

We conduct experiments on 2 human preference datasets: HHH Alignment (Askell et al., 2021) and MT Bench Human Judgment (Zheng et al., 2023) that use a ranking grading scheme. In Table 4, results show that prompting Llama-2-Chat surprisingly obtains reasonable performance, which we conjecture might be the effect of using a base model that is trained with Reinforcement Learning from Human Feedback (RLHF). When training on feedback derived from coarse-grained score rubrics (denoted as Llama2-Chat 13B + Coarse), it only hurts performance. On the other hand, Prometheus 13B shows a +5.43% and +5.38% margin over its base model Llama2-Chat-13B on the HHH Alignment and MT Bench Human Judgement dataset, respectively. These results are surprising because they indicate that training on an absolute grading scheme could also improve performance on a ranking grading scheme even without directly training on ranking evaluation instances. Moreover, it shows the possibilities of using a generative LLM (Prometheus) as a reward model for RLHF (Kim et al., 2023b). We leave the exploration of this research to future work.

## 7 Conclusion

In this paper, we discuss the possibility of obtaining an open-source LM that is specialized for fine-grained evaluation. While text evaluation is an inherently difficult task that requires multi-faceted considerations, we show that by incorporating the appropriate reference material, we can effectively induce evaluation capability into an LM. We propose a new dataset called the Feedback Collection that encompasses thousands of customized score rubrics and train an open-source evaluator model, Prometheus. Surprisingly, when comparing the correlation with human evaluators, Prometheus obtains a Pearson correlation on par with GPT-4, while the quality of the feedback was preferred over GPT-4 58.62% of the time. When comparing Pearson correlation with GPT-4, Prometheus shows the highest correlation even outperforming GPT-3.5-Turbo. Lastly, we show that Prometheus shows superior performance on human preference datasets, indicating its possibility as an universal reward model. We hope that our work could stimulate future work on using open-source LLMs as evaluators instead of *solely* relying on proprietary LLMs.

ACKNOWLEDGMENTS

This work was partly supported by KAIST-NAVER Hypercreative AI Center and Institute of Information & communications Technology Planning & Evaluation (IITP) grant funded by the Korea government (MSIT) (No.2022-0-00264, Comprehensive Video Understanding and Generation with Knowledge-based Deep Logic Neural Network, 40%; No.2021-0-02068, Artificial Intelligence Innovation Hub, 20%). We thank Minkyeong Moon, Geonwoo Kim, Minkyeong Cho, Yerim Kim, Sora Lee, Seunghwan Lim, Jinheon Lee, Minji Kim, and Hyorin Lee for helping with the human evaluation experiments. We thank Se June Joo, Dongkeun Yoon, Doyoung Kim, Seonghyeon Ye, Gichang Lee, and Yehbin Lee for helpful feedback and discussions.

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

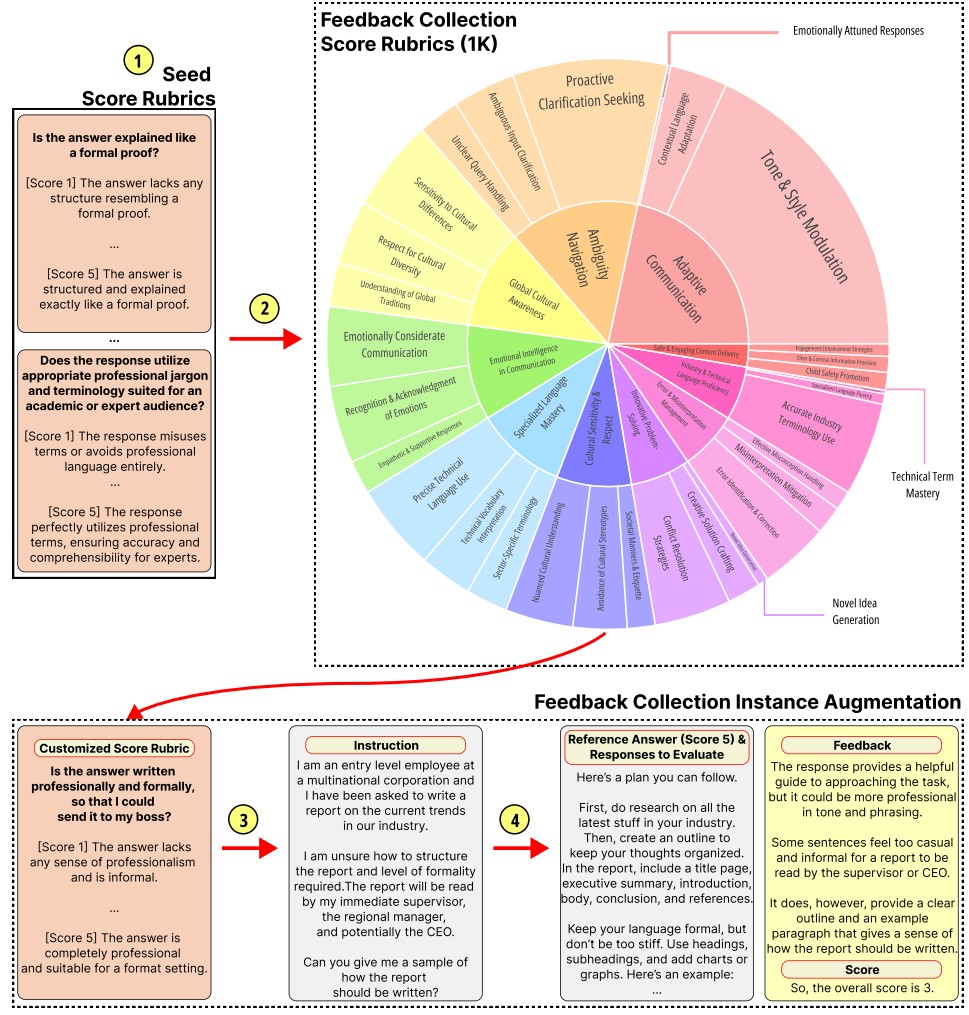

Figure 6: Overview of the augmentation process of the FEEDBACK COLLECTION. The keywords included within the score rubrics of the FEEDBACK COLLECTION is also displayed.

## A  OVERALL ILLUSTRATION OF THE DATA AUGMENTATION PROCESS

Figure 6 shows the augmentation process of the FEEDBACK COLLECTION.

## B  EXTENSION OF EXPERIMENTAL SETTING: EVALUATING AN EVALUATOR LLM

Detailed information on the datasets used for the experiment is included in Table 5.

### B.1  LIST OF EXPERIMENTS AND METRICS

**Absolute Grading**   We first test in an Absolute Grading setting, where the evaluator LM should generate a feedback and score within the range of 1 to 5 given an instruction, a response to evaluate, and reference materials (as shown in Figure 2). Absolute Grading is challenging compared to Ranking Grading since the evaluator LM does not have access to an opponent to compare with and it is required to provide a score *solely* based on its internal decision. Yet, it is more practical for users since it relieves the need to prepare an opponent to compare with during evaluation.

Table 5: Information about the datasets we use to test evaluator LMs. Note that FEEDBACK BENCH is a dataset that is crafted with the exact same procedure as the FEEDBACK COLLECTION as explained in Section 3.1. We include additional analysis of FEEDBACK BENCH in Appendix E. Simulated GPT-4 † denotes GPT-4 prompted to write a score of $i$ ($1 \leq i \leq 5$) during augmentation.

| Evaluation Mode | Evaluation Data | Source / Types of Score Rubric | Response LMs |
|---|---|---|---|
| **Absolute Evaluation** | FEEDBACK BENCH (Seen Rubric) | 1K Machine Generated | Simluated GPT-4 † |
| | FEEDBACK BENCH (Unseen Rubric) | 50 Hand Crafted | |
| | Vicuna Bench | 80 Hand Crafted | WizardLM, Vicuna, Llama2-Chat, ChatGPT |
| | MT Bench | | |
| | Flask Eval | Logical Thinking (3), Background Knowledge (2) Problem Handling (4), User Alignment (3) | Alpaca, Vicuna, Bard, ChatGPT |
| **Ranking Evaluation** | MT Bench Human Judgments | Human Preference (Helpfulness) | Alpaca, Llama, Vicuna, ChatGPT, Claude-v1, GPT-4 |
| | HHH Alignment | Helpfulness, Harmlessness, Honesty, Other | Human Annotation |

We mainly conduct three experiments in this setting: (1) measuring the correlation with human evaluators (Section 5.1), (2) comparing the quality of the feedback using human evaluation (Section 5.1), and (3) measuring the correlation with GPT-4 evaluation (Section 5.2). For the experiments that measure the correlation, we use 3 different correlation metrics: **Pearson**, **Kdendall-Tau**, and **Spearman**. For measuring the quality of the generated feedback, we conduct a **pairwise comparison** between the feedback generated by PROMETHEUS, GPT-3.5-Turbo, and GPT-4, asking human evaluators to choose *which* has better quality and *why* they thought so. Specifically, we recruited 9 crowdsource workers and split them into three groups: PROMETHEUS vs GPT-4, PROMETHEUS vs ChatGPT, and GPT-4 vs ChatGPT. The annotators are asked to answer the following three questions:

1. What score would you give to the response based on the given score rubric?
2. Among the two Feedback, which is better for critiquing the given response?
3. Why did you reject that particular feedback?

We use the following four benchmarks to measure the correlation with human evaluation and GPT-4 evaluation. Note that FEEDBACK BENCH is a dataset generated with the same procedure as the FEEDBACK COLLECTION, and is divided into two subsets (Seen Rubric and Unseen Rubric).

- **FEEDBACK BENCH**: The **Seen Rubric** subset shares the same 1K score rubrics with the FEEDBACK COLLECTION across 1K instructions (1 per score rubric). The **Unseen Rubric** subset also consists of 1K new instructions but with 50 new score rubrics that are generated the same way as the training set. Details are included in Appendix E.
- **Vicuna Bench**: We adapt the 80 test prompt set from Vicuna (Chiang et al., 2023) and hand-craft customized score rubrics for each test prompt. In order to obtain reference answers, we concatenate the hand-crafted score rubric and instruction to prompt GPT-4.
- **MT Bench**: We adapt the 80 test prompt set from MT Bench (Zheng et al., 2023), a multi-turn instruction dataset. We hand-craft customized score rubrics and generate a reference answer using GPT-4 for each test prompt as well. Note that we only use the last turn of this dataset for evaluation, providing the previous dialogue as input to the evaluator LM.
- **FLASK Eval**: We adapt the 200 test prompt set from FLASK (Ye et al., 2023b), a fine-grained evaluation dataset that includes multiple conventional NLP datasets and instruction datasets. We use the 12 score rubrics (that are relatively coarse-grained compared to the 1K score rubrics used in the FEEDBACK COLLECTION) such as Logical Thinking, Background Knowledge, Problem Handling, and User Alignment.

**Ranking Grading** To test if an evaluator LM trained only on Absolute Grading could be utilized as a universal reward model based on *any* criteria, we use existing human preference benchmarks and use **accuracy** as our metric. Specifically, we check whether the evaluator LM could give a higher score to the response that is preferred by human evaluators. The biggest challenge of employing an

evaluator LM trained in an Absolute Grading setting and testing it on Ranking Grading was that it could give the same score for both candidates. Therefore, we use a temperature of 1.0 when evaluating each candidate independently and iterate until there is a winner. Hence, it's noteworthy that the settings are not exactly fair compared to other ranking models. This setting is NOT designed to claim SOTA position in these benchmarks, but is conducted only for the purpose of checking whether an evaluator LM trained in an Absolute Grading setting could also generalize in a Ranking Grading setting according to *general* human preference. Also, in this setting, we do not provide a reference answer to check whether PROMETHEUS could function as a reward model. We use the following two benchmarks to measure the accuracy with human preference datasets:

- **MT Bench Human Judgement**: This data is another version of the aforementioned MT Bench (Zheng et al., 2023). Note that it includes a tie option as well and does not require iterative inference to obtain a clear winner. We use Human Preference as our criteria.
- **HHH Alignment**: Introduced by Anthropic (Askell et al., 2021), this dataset (221 pairs) is one of the most widely chosen reward-model test-beds that measures preference accuracy in Helpfulness, Harmlessness, Honesty, and in General (Other) among two response choices.

## B.2 BASELINES

The following list shows the baselines we used for comparison in the experiments:

- LLAMA2-CHAT-{7,13,70}B (Touvron et al., 2023): The base model of PROMETHEUS when fine-tuning on the FEEDBACK COLLECTION. Also, it is considered as the best option among the open-source LLMs, which we use as an evaluator in this work.
- LLAMA-2-CHAT-13B + COARSE: To analyze the effectiveness of training on thousands of fine-grained score rubrics, we train a comparing model only using 12 coarse-grained score rubrics from Ye et al. (2023b). Detailed information on this model is in Appendix G.
- GPT-3.5-TURBO-0613: Proprietary LLM that offers a cheaper price when employed as an evaluator LLM. While it is relatively inexpensive compared to GPT-4, it still has the issue of uncontrolled versioning and close-source nature.
- GPT-4-{0314,0613, RECENT}: One of the most powerful proprietary LLM that is considered the main option when using LLMs as evaluators. Despite its reliability as an evaluator LM due to its superior performance, it has several issues of prohibitive costs, uncontrolled versioning, and close-source nature.
- STANFORDNLP REWARD MODEL[2]: One of the state-of-the-art reward model directly trained on multiple human preference datasets in a ranking grading setting.
- ALMOST REWARD MODEL (Kim et al., 2023b): Another state-of-the-art reward model trained on synthetic preference datasets in a ranking grading setting.

## C EXTENSION OF DISCUSSIONS AND ANALYSIS

### C.1 WHY IS IT IMPORTANT TO INCLUDE REFERENCE MATERIALS?

Evaluating a given response without any reference material is a very challenging task (i.e., Directly asking to decide a score only when an instruction and response are given), since the evaluation LM should be able to (1) know what the important aspects tailored with the instruction is, (2) internally estimate what the answer of the instruction might be, and (3) assess the quality of responses based on the information derived from the previous two steps. Our intuition is that by incorporating each component within the reference material, the evaluator LM could solely focus on assessing the quality of the response instead of determining the important aspects or solving the instruction. Specifically, we analyze the role of each component as follows:

- **Score Rubric**: Giving information of the the pivotal aspects essential for addressing the given instruction. Without the score rubric, the evaluator LM should inherently know what details should be considered from the given instruction.

---

[2]https://huggingface.co/stanfordnlp/SteamSHP-flan-t5-xl

Table 6: Pearson, Kendall-Tau, Spearman correlation with data generated by GPT-4-0613 (Feedback Collection Test set) and scores sampled from GPT-4-0613 across 3 inferences (Vicuna Bench).

| Evaluator LM | FEEDBACK COLLECTION TEST set | | VICUNA BENCH |
| | Seen Score Rubric | Unseen Score Rubric | - |
| | Pearson | Pearson | Pearson |
|---|---|---|---|
| PROMETHEUS 7B | 0.860 | 0.847 | 0.457 |
| **Training Ablation** | | | |
| W/O SCORE RUBRIC | 0.837 | 0.745 | 0.355 |
| W/O FEEDBACK DISTILLATION | 0.668 | 0.673 | 0.413 |
| W/O REFERENCE ANSWER | 0.642 | 0.626 | 0.349 |
| **Model Ablation** | | | |
| LLAMA-2 7B BASELINE | 0.839 | 0.818 | 0.404 |
| VICUNA-v1.5 7B BASELINE | 0.860 | 0.829 | 0.430 |
| CODE-LLAMA 7B BASELINE | 0.823 | 0.761 | 0.470 |

- **Reference Answer**: Decomposing the process of estimating a reference answer and evaluating it at the same time into two steps. Since the reference answer is given as an additional input, the evaluator LM could only focus on evaluating the given response. This enables to bypass a natural proposition that if an evaluator LM doesn't have the ability to solve the problem, it's likely that it cannot evaluate different responses effectively as well.

As shown in Table 6, we conduct an ablation experiment by excluding each reference material and also training only on the score rubric without generating a feedback. Additionally, we also ablate the effect of using different model variants (Llama-2, Vicuna, Code-Llama) instead of Llama-2-Chat.

## C.2 ABLATION EXPERIMENTS

**Training Ablation** The results indicate that each component contributes orthogonally to PROMETHEUS's superior evaluation performance. Especially, excluding the reference answer shows the most significant amount of performance degradation, supporting our claim that including a reference answer relieves the need for the evaluator LM to internally solve the instruction and only focus on assessing the response. Also, while excluding the score rubric on the FEEDBACK BENCH does not harm performance a lot, the performance drops a lot when evaluating on Vicuna Bench. As in our hypothesis, we conjecture that in order to generalize on other datasets, the role of providing what aspect to evaluate holds relatively crucial.

**Model Ablation** To test the effect using LLAMA2-CHAT, a model that has been instruction-tuned with both supervised fine-tuning and RLHF, we ablate by using different models as a starting point. Results show that different model choices do not harm performance significantly, yet a model trained with both supervised fine-tuning and RLHF shows the best performance, possibly due to additional training to follow instructions. However, we find that using Code-Llama has some benefits when evaluating on code domain, and we discuss the effect on Section C.6.

## C.3 NARROWING PERFORMANCE GAP TO GPT-4 EVALUATION

The observed outcomes, in which PROMETHEUS consistently surpasses GPT-4 based on human evaluations encompassing both scores and quality of feedback, as well as correlations in the FEEDBACK BENCH, are indeed noteworthy. We firmly posit that these findings are not merely serendipitous and offer the following justifications:

- Regarding results on FEEDBACK BENCH, our model is directly fine-tuned on this data, so it's natural to beat GPT-4 on a similar distribution test set if it is well-trained. In addition, for GPT-4, we compare the outputs of **inferencing** on the instructions and **augmenting** new instances, causing the self-consistency to be lower.

- Regarding score correlation for human evaluation, our model shows similar or slightly higher trends. First, our human evaluation set-up excluded all coding or math-related questions, which is where it is non-trivial to beat GPT-4 yet. Secondly, there's always the margin of human error that needs to be accounted for. Nonetheless, we highlight that we are the first work to argue that an open-source evaluator LM could closely reach GPT-4 evaluation *only when the appropriate reference materials are accompanied.*

- As shown in Figure 5, PROMETHEUS tends to be critical compared to GPT-4. We conjecture this is because since it is specialized for evaluation, it acquires the characteristics of seeking for improvement when assessing responses.

### C.4 QUALITATIVE EXAMPLES OF FEEDBACK GENERATED BY PROMETHEUS

We present five qualitative examples to compare the feedback generated by PROMETHEUS and GPT-4 in Appendix L. Specifically, Figure 16 shows an example where human annotators labeled that GPT-4 generate an abstract/general feedback not suitable for criticizing the response. Figure 17 shows an example where human annotators labeled that PROMETHEUS generate overly critical feedback. Figure 18 shows an example of human annotators labeled as a tie. In general, PROMETHEUS generates a detailed feedback criticizing *which* component within the response is wrong and seek improvement. This qualitatively shows that PROMETHEUS could function as an evaluator LM.

Moreover, we present an example of evaluating python code responses using PROMETHEUS, GPT-4, and Code-Llama in Figure 19. We discuss the effect of using a base model specialized on code domain for code evaluation in Section C.6.

### C.5 A PRACTITIONER'S GUIDE FOR DIRECTLY USING PROMETHEUS EVALUATION

**Preparing an Evaluation Dataset**   As shown in the previous sections, PROMETHEUS functions as a good evaluator LM not only on the FEEDBACK BENCH (a dataset that has a similar distribution with the dataset it was trained on), but also on other widely used evaluation datasets such as the Vicuna Bench, MT Bench, and Flask Eval. As shown in Figure 1, users should prepare the instruction dataset they wish to evaluate their target LLM on. This could either be a widely used instruction dataset or a custom evaluation users might have.

**Deciding a Score Rubric to Evaluate on**   The next step is to choose the score rubric users would want to test their target LLM on. This could be confined to generic metrics such as helpfulness/harmlessness, but PROMETHEUS also supports fine-grained score rubrics such as "Child-Safety", "Creativity" or even "Is the response formal enough to send to my boss".

**Preparing Reference Answers**   While evaluating without any reference is also possible, as shown in Table 6, PROMETHEUS shows superior performance when the reference answer is provided. Therefore, users should prepare the reference answer they might consider most appropriate based on the instructions and score rubrics they would want to test on. While this might require additional cost to prepare, there is a clear trade-off in order to improve the precision or accuracy of the overall evaluation process, hence it holds crucial.

**Generating Responses using the Target LLM**   The last step is to prepare responses acquired from the target LLM that users might want to evaluate. By providing the reference materials (score rubrics, reference answers) along with the instruction and responses to evaluate on, PROMETHEUS generates a feedback and a score. Users could use the score to determine how their target LLM is performing based on customized criteria and also refer to the feedback to analyze and check the properties and behaviors of their target LLM. For instance, PROMETHEUS could be used as a good alternative for GPT-4 evaluation while training a new LLM. Specifically, the field has not yet come up with a formalized procedure to decide the details of instruction-tuning or RLHF while developing a new LLM. This includes deciding how many training instances to use, how to systematically decide

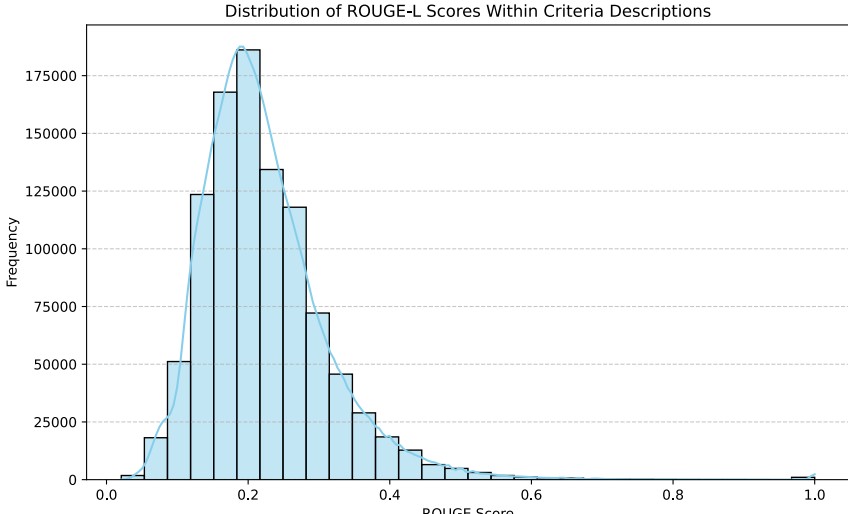

Figure 7: Rouge-L score distribution among two randomly sampled score rubrics from the FEED-BACK COLLECTION. A left-skewed distribution with low values shows the score rubrics are diverse.

the training hyperparameters, and quantitatively analyzing the behaviors of LLMs across multiple versions. Most importantly, users might not want to send the outputs generated by their LLMs to OpenAI API calls. In this regard, PROMETHEUS provides an appealing solution of having control over the whole evaluation process, also supporting customized score rubrics.

### C.6 A PRACTITIONER'S GUIDE FOR TRAINING A NEW EVALUATION MODEL

Users might also want to train their customized evaluator LM as PROMETHEUS for different use cases. As shown in Table 3, training directly on the Flask dataset (denoted as LLAMA2-CHAT 13B + COARSE) shows a higher correlation with GPT-4 on the Flask Eval dataset compared to PROMETHEUS that is trained on the FEEDBACK COLLECTION. This implies that directly training on a target feedback dataset holds the best performance when evaluating on it. Yet, this requires going through the process of preparing a new feedback dataset (described in Section 3.1). This implies that there is a trade-off between obtaining a strong evaluator LM on a target task and paying the initial cost to prepare a new feedback dataset. In this subsection, we provide some guidelines for how users could also train their evaluator LM using feedback datasets.

**Preparing a Feedback Dataset to train on**    As described in Section 3, some important considerations to prepare a new feedback dataset are: (1) including as many reference materials as possible, (2) maintaining a uniform length among the reference answers for each score (1 to 5) to prevent undesired length bias, (3) maintaining a uniform score distribution to prevent undesired decision bias. While we did not explore the effect of including other possible reference materials such as a "Score 1 Reference Answer" or "Background Knowledge" due to limited context length, future work could also explore this aspect. The main intuition is that providing more reference materials could enable the evaluator LM to solely focus on evaluation instead of solving the instruction.

**Choosing a Base Model to Train an Evaluator LM**    As shown in Figure 19, we find that training on CODE-LLAMA provides more detailed feedback and a reasonable score decision when evaluating responses on code domains (7 instances included within the Vicuna Bench dataset). This indicates that choosing a different base model based on the domain to evaluate might be crucial when designing an evaluator LM. We also leave the exploration of training an evaluator LM specialized on different domains (e.g., code and math) as future work.

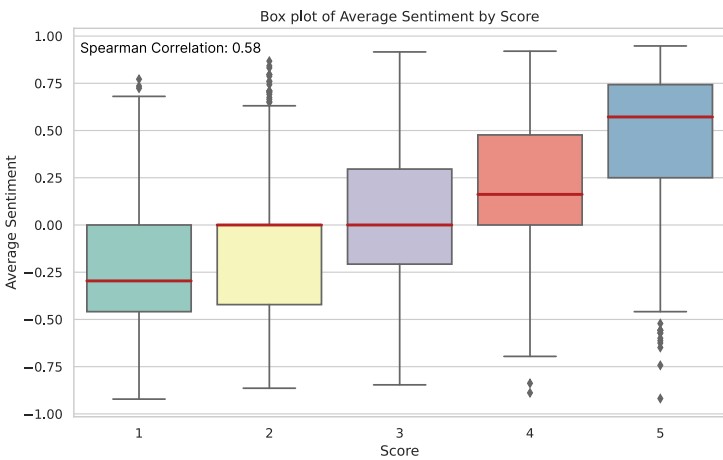

Figure 8: Box and whisker plot for average sentiment per score description. A linearly increasing trend is crucial for the evaluator LM to decide a score in an Absolute Scoring setting.

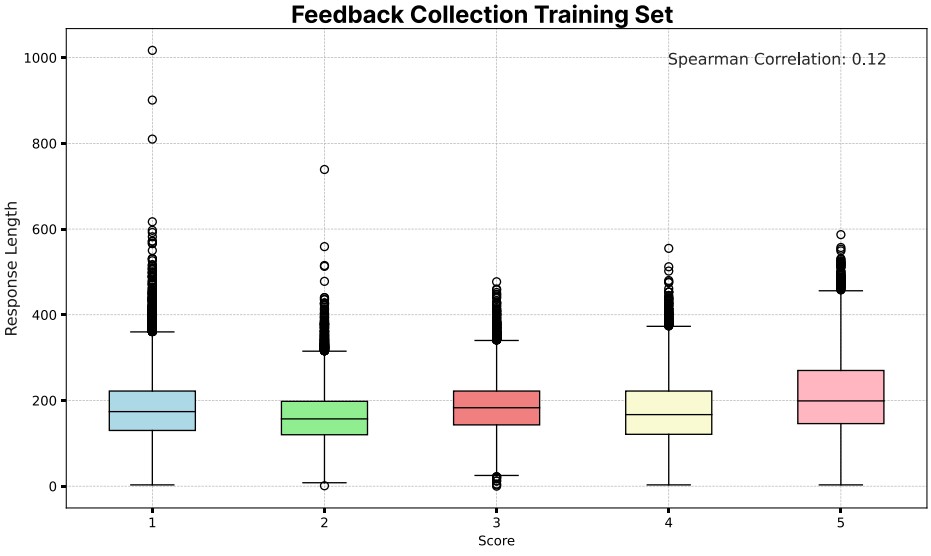

Figure 9: Box and whisker plot plotting the length distribution of responses on each score range. We check whether there is a length bias (i.e., a higher score given for longer responses).

## D ANALYSIS OF THE FEEDBACK COLLECTION DATASET

In this section, we provide a comprehensive analysis of the characteristics of the FEEDBACK COL-LECTION dataset. To ensure the quality, we answer each question one by one, emphasizing our main considerations during the creation of the dataset.

**Are the Score Criteria Diverse Enough?**    Following previous work (Wang et al., 2022; Honovich et al., 2022), we plot the rouge-L distribution between two instances among our whole set of 1K score rubrics. Specifically, we use the score criteria (description of the criteria) and measure the rouge-L value between the two score criteria. Figure 7 shows the overall distribution plot. The results indicate that each criteria does not overlap with one another, ensuring that we include many *novel* score rubrics in our training set.

Table 7: Distinct N-gram measured on each component of the training instance. A higher diversity ratio indicates that each component tends to be more diverse.

| Distinct N-gram | Bigram Diversity Ratio | Trigram Diversity Ratio |
|---|---|---|
| Instruction | 0.43 | 0.79 |
| Reference | 0.43 | 0.82 |
| Score Rubric | 0.60 | 0.81 |
| Responses | 0.32 | 0.77 |
| Feedback | 0.26 | 0.66 |

**Are the Score Descriptions Well Formulated?** Another component in the score rubric is a description of each score (i.e., A comprehensive reason why a score of $i$ ($1 \leq i \leq 5$ should be given). In an Absolute Scoring setting, it is important to evaluate the given response based on the score descriptions instead of giving a score of 1 for all responses that lack a minor detail or giving a score of 5 for all responses that seem to be good on the surface. Due to these reasons, the role of the score descriptions hold crucial, where the main role is to show a monotonically increasing tendency of sentiment, not dramatically. Figure 8 shows that the FEEDBACK COLLECTION holds a smoothly increasing sentiment tendency for each score description. This ensures the quality of the score rubric, confirming that it plays a role in deciding the score.

**Is there a length bias among the Responses?** Previous work has demonstrated that when LMs are used as evaluators, they tend to give higher scores to longer responses (Li et al., 2023; Dubois et al., 2023; Zheng et al., 2023). In order to minimize this effect during fine-tuning PROMETHEUS, one of our main consideration was to maintain a length distribution equal among the score range of 1 to 5. As shown in Figure 9, most of the responses within the FEEDBACK COLLECTION maintained a similar length among different scores (near 200 tokens). We also include a comprehensive analysis of whether PROMETHEUS possessed any length bias during evaluation in Appendix I.

**Are the Instructions, Responses, and Feedback Diverse as Well?** In addition to the analysis of the score rubric and responses, we also analyze whether the instructions, responses, and feedback within the FEEDBACK COLLECTION are diverse enough. For this purpose, we examine the bigram and trigram ratios. The results are shown in Table 7, indicating a variety in how terms are expressed, and our findings suggest a moderate level of diversity. While there is some term repetition, the dataset also showcases a notable range of expressions.

## E    ANALYSIS OF THE FEEDBACK BENCH EVALUATION DATASET

In this section, we provide a analysis of whether the FEEDBACK BENCH consists of unseen score rubrics against the score rubrics from the FEEDBACK COLLECTION.

**Does the testset maintain Unseen Score Rubrics?** One of the main considerations of our experiments in Section 5.2 using the FEEDBACK BENCH was testing whether PROMETHEUS could generalize to unseen *customized* score rubrics. For this purpose, we built an unseen customized rubric subset. We plot the rouge-L distribution between a random score rubric within the FEEDBACK COLLECTION and a random score rubric within the FEEDBACK BENCH. As shown in Figure 10, there is a low overlap among the train and test sets, confirming that the FEEDBACK BENCH is valid to be claimed as an *unseen* test set to measure the evaluation capability of evaluator LMs.

## F    FINE-TUNING AND INFERENCE DETAILS OF PROMETHEUS

We use 8xA100 (80GB) GPUs to train our models with PyTorch Fully-Sharded Data Parallel (FSDP) option. The code we used for training and inference is the official Llama2 fine-tuning code released by Meta AI[3]. The hyper-parameters we used are the basic settings in the fine-tuning code except

---

[3]https://github.com/facebookresearch/llama-recipes

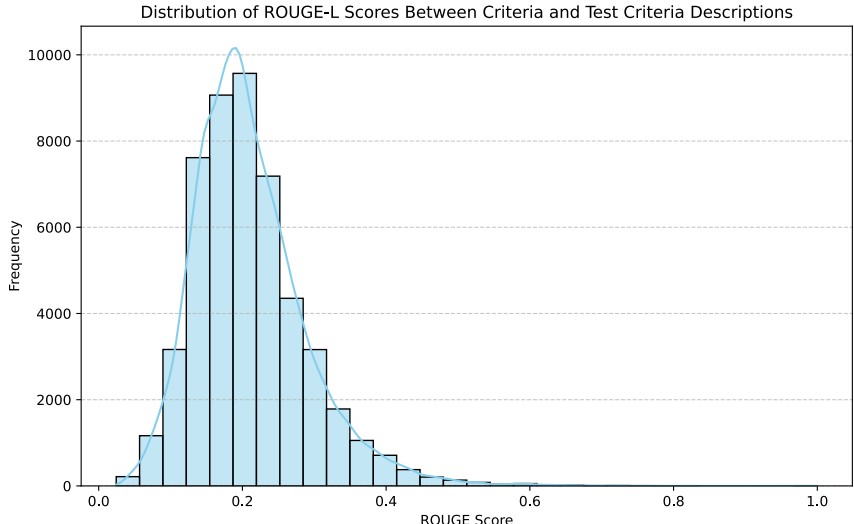

Figure 10: Rouge-L score distribution among a randomly sampled score rubric from the FEEDBACK COLLECTION and a score rubric from the FEEDBACK BENCH. A left-skewed distribution with low values shows that they do not overlap with each other, hence meaning that a UNSEEN score rubric assumption is satisfied.

Table 8: Hyperparameters used for fine-tuning PROMETHEUS.

| Model | Base Model | Batch size | LR | LR Scheduler | Optimizer | Max Length (Input & Output) |
|---|---|---|---|---|---|---|
| PROMETHEUS-7B | Llama-2-Chat-7B | 28 | 1e-5 | StepLR | AdamW | 4096 |
| PROMETHEUS-13B | Llama-2-Chat-13B | 20 | 1e-5 | StepLR | AdamW | 4096 |

Table 9: Hyperparameters used for inferencing PROMETHEUS, GPT-3.5-Turbo, and GPT-4. Verbalizer denotes accepting outputs such as "[Score 5]" or "Score: 4 out of 5" whereas the exact format is "[Result] 5" (format is mentioned concretely within the instruction given to the evaluator LM). Even after applying a verbalizer, Llama-2-Chat is not able to generate a score decision that could easily be parsed, highlighting the benefits of fine-tuning it on feedback data.

| Params | Model | Temperature | Top-p | Repetition Penalty | Max Output Length | Verbalizer |
|---|---|---|---|---|---|---|
| 7B | Llama-2-Chat-7B | 1.0 | 0.9 | 1.03 | 1024 | Yes |
| 13B | Llama-2-Chat-13B | 1.0 | 0.9 | 1.03 | 1024 | Yes |
| 70B | Llama-2-Chat-70B | 1.0 | 0.9 | 1.03 | 1024 | Yes |
| 7B | PROMETHEUS-7B | 1.0 | 0.9 | 1.03 | 1024 | No |
| 13B | PROMETHEUS-13B | 1.0 | 0.9 | 1.03 | 1024 | No |
| - | GPT-3.5-Turbo | 1.0 | 0.9 | - | 1024 | No |
| - | GPT-4 | 1.0 | 0.9 | - | 1024 | No |

for the training batch size which was set according to the model size: for 7B models we used 28 and for 13B models we used 20 to fully leverage GPU memory. Note that in the official Llama2 fine-tuning code, the loss is only calculated on the feedback and score decision, not the instruction. We empirically find that not masking out the instruction leads to poor performance while evaluating responses. The detailed hyper-parameters are shown in Table 8.

For inference, we use the hyper-parameters as shown in Table 9. When inferencing with the naive Llama-2-Chat model (not trained on the FEEDBACK COLLECTION) it was extremely difficult to steer the model to generate a final score in the form to be easily parsed (e.g., "[RESULT] 3"). While in-context learning (ICL) could solve this issue, most of our instances contained a maximum of 3072 tokens, so we could not utilize demonstrations during inference. Therefore, we empirically

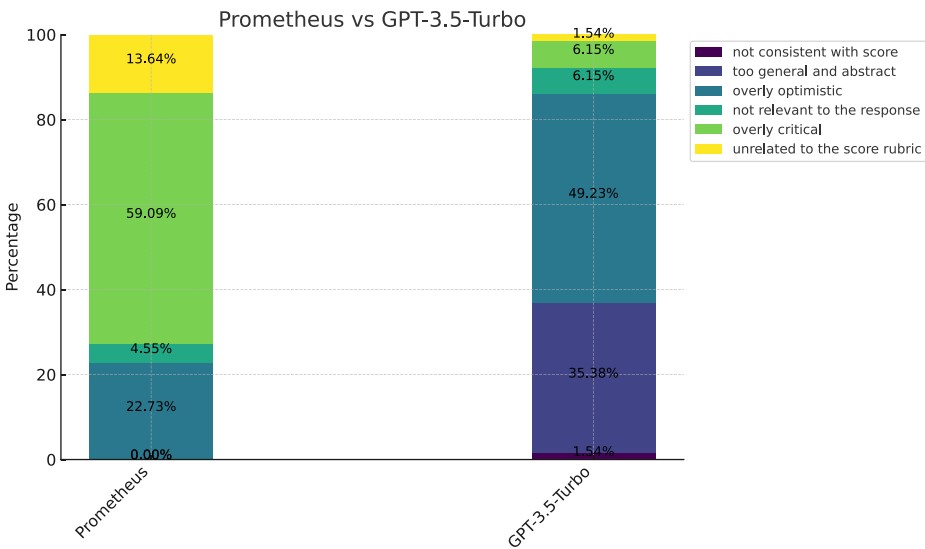

Figure 11: The reason why GPT-3.5-Turbo's or Prometheus's feedback was not chosen over the other. PROMETHEUS generates less abstract and general feedback, but tends to write overly critical ones.

found patterns such as "[SCORE 5]" or "Score: 4 out of 5" and applied verbalizer to map those outputs to a final score decision. This highlights the benefits of directly training to generate in a structured format as PROMETHEUS. On the other hand, we also find that proprietary LLMs such as GPT-3.5-Turbo and GPT-4 excel at generating structured outputs when the prompt is adeptly given. Also, note that we found that if we set the temperature to 0.0, evaluator LMs are not able to generate meaningful feedback compared to using a temperature of 1.0.

## G  TRAINING A EVALUATOR LM ON COARSE-GRAINED SCORE RUBRICS

For the purpose of exploring the benefits of training on thousands of fine-grained and customized score rubrics, we employ a baseline of only training on relatively *coarse-grained* score rubrics. Since the FEEDBACK COLLECTION's instructions are closely tied with the score rubrics during its creation process, we could not directly use it and only change the score rubrics into coarse-grained ones.

So, we used the Flask dataset (Ye et al., 2023b) and split it into training data and evaluation data. The evaluation data is denoted as Flask Eval throughout the paper. Specifically, the Flask dataset consists of 1.7K instructions acquired across conventional NLP datasets and instruction datasets. Also, there exists 76.5K responses acquired across 15 response LMs. Each instance has a score rubric among 12 options (Logical Robustness, Logical Correctness, Logical Efficiency, Factuality, Commonsense Understanding, Harmlessness, Readability, Comprehension, Insightfulness, Completeness, Metacognition, Conciseness). While these 12 score rubrics are more fine-grained and diverse compared to previous works only using helpfulness and harmlessness, they are coarse-grained compared to the thousands of score rubrics included within the FEEDBACK COLLECTION, so we denote as coarse-grained in this work.

Among the 1.5K instructions & 67.5K responses as training data, we found that the score distribution is extremely skewed towards the score of 5. We distributed the instances so that the number of instances within the score range of 1 to 5 remains equal, which leads to 30K training instances. We trained the Llama-2-Chat model on the Flask train set, which led to one of our baselines denoted as LLAMA-2-CHAT + COARSE in Table 2, Table 3, Table 4, Table 6.

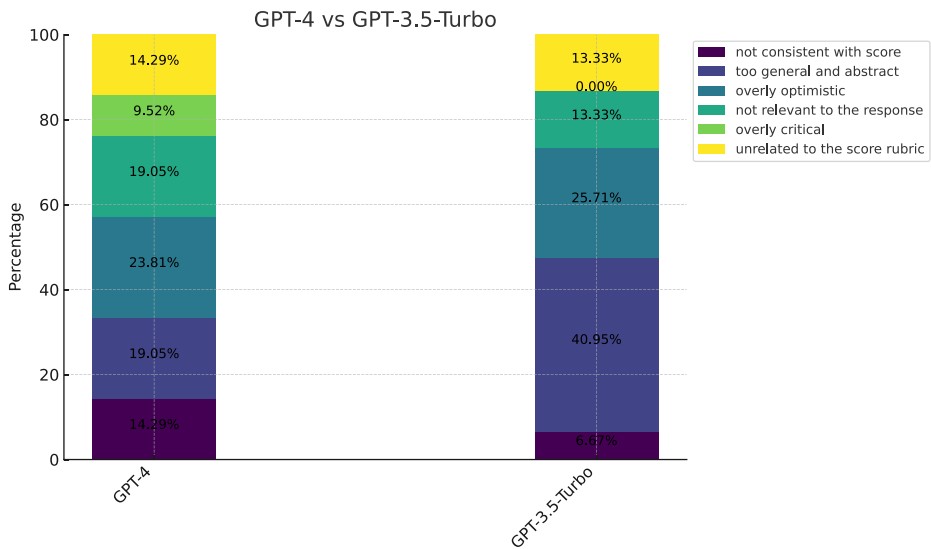

Figure 12: The reason why GPT-4's or GPT-3.5-Turbo's feedback was not chosen over the other. GPT-4 generates less abstract and general feedback, but tends to write overly critical ones.

## H  PAIRWISE COMPARISON OF THE QUALITY OF THE FEEDBACK

In this section, we further explain the experimental setting and present additional results & analysis for the experiment of comparing the quality of the generated feedback (Section 5.1).

In addition to Figure 5, the reason *why* each annotator rejected the feedback from either PROMETHEUS, GPT-3.5-Turbo, GPT-4 is shown in Figure 11 and Figure 12.

The results further support our claim that PROMETHEUS tends to be critical over GPT-4 and GPT-3.5-Turbo. Interestingly, GPT-4 was considered to be more critical compared to GPT-3.5-Turbo and the gap was even wider when comparing GPT-3.5-Turbo and PROMETHEUS. This indicates that PROMETHEUS can serve as a critical judge when evaluating responses generated by LLMs, but it could also be biased towards not being optimistic generally. The degree of being critical could be useful or a limitation based on different use cases. For instance, we conjecture that it could be helpful when analyzing the limitations of LLMs or providing feedback as supervision to further improve a target LLM (e.g., RLHF), yet we leave this exploration to future work.

## I  IS THERE A LENGTH BIAS DURING EVALUATION?

One of the limitations of employing an LLM as an evaluator LM is that it could be vulnerable to various biases. In this work, we train/test on an Absolute Scoring evaluation setting, hence there exists no position bias. Yet, it is crucial to analyze whether PROMETHEUS showed any bias towards favoring longer responses. Hence, we conduct a comprehensive analysis in this section.

As shown in Figure 13, Figure 14, and Figure 15, both GPT-4 and PROMETHEUS and GPT-4 shows a similar trend of not favoring longer responses (i.e., similar length distribution among different scores). However, as mentioned in Zheng et al. (2023), LLM evaluators might favor more verbose responses, yet the responses from our test instances (FEEDBACK BENCH and Vicuna Bench) did not include any adversarial examples to test this phenomenon. More extensive research on whether the length bias is also transferred to fine-tuned evaluator LMs should be explored in future work.

## J  PROMPT FOR FEEDBACK COLLECTION CREATION

In this section, we provide the extensive list of prompts used to create the FEEDBACK COLLECTION.

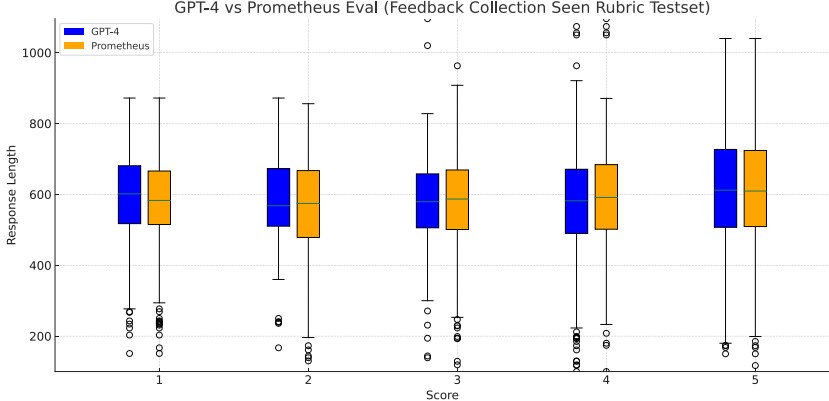

Figure 13: Box and whisker plot describing a relationship between a given response and its corresponding score. We check if the response lengths correlate with its scores.

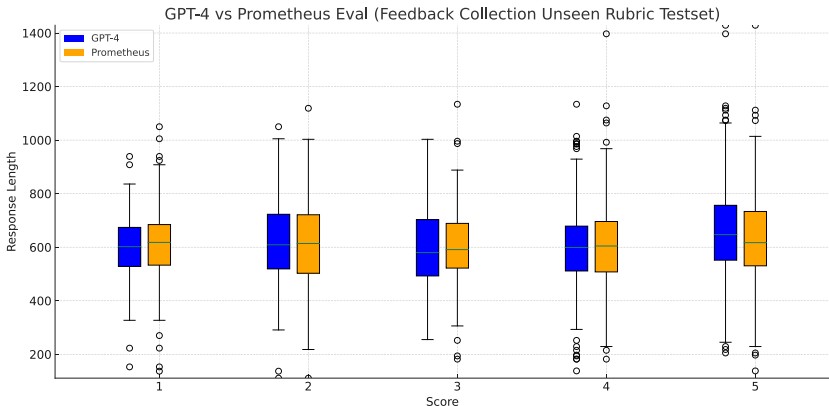

Figure 14: Box and whisker plot describing a relationship between a given response and its corresponding score. We check if the response lengths correlate with its scores.

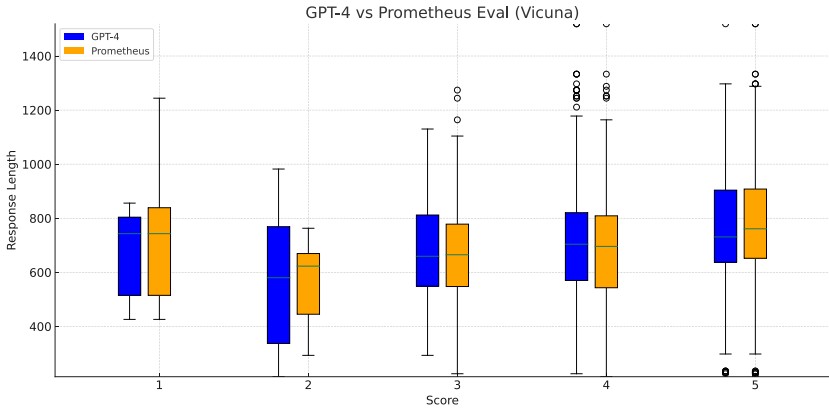

Figure 15: Box and whisker plot describing a relationship between a given response and its corresponding score. We check if the response lengths correlate with its scores.

Note that in the prompt of generating a response and a feedback, we use the sentence length of the reference answer and append it to "{SENT NUM}" within the prompt. This was crucial to make the length even across different scores as shown in Figure 9. Also, note that for the 1K

score rubrics, 20K instructions & reference answers, and 100K responses & feedback within the FEEDBACK COLLECTION, each prompt was sequentially used. In early experiments, we found that generating every component all at once leads to very poor generation quality (i.e., similar responses & feedback across different score ranges). Yet, we found that grouping (1) the instruction and reference answer generation and (2) the response and feedback generation had a positive synergy, leading to better generation quality and less amount of cost. Also, to the best of our knowledge, we are first to explore acquiring negative and neutral responses (Score $1 \sim 4$ responses) through GPT-4 augmentation. We hope future work could also explore applying this strategy to different use cases.

---

**Prompt for Brainstorming New Score Rubrics**

We are brainstorming criteria with which to grade a language model on its responses in diverse situations.
A 'criteria' is some useful, real-world objective, and associated rubric for scores 1-5, that tests a capability.

Here you will see 4 examples of 'criteria', and their scoring rubrics, formatted as JSON.
Criteria 1:
{JSON LIST 1}

Criteria 2:
{JSON LIST 2}

Criteria 3:
{JSON LIST 3}

Criteria 4:
{JSON LIST 4}

Please brainstorm a new criteria and scoring rubrics.
Be creative and create new but useful criteria that people in different settings or industries might find practical.
Please format the output as same as the above examples with no extra or surrounding text.
Write [END] after you are done.

New Criteria:

---

**Prompt for Paraphrasing as a New Score Rubric**

Please paraphrase the sentences inside the dictionary below.
Each paraphrase should not change the meaning or substance of the original sentence, be naturally written, but sufficiently diverse from one another.
Diversity can come from differences in diction, phrasing, sentence structure, formality, detail, and/or other stylistic changes.

The dictionary:
{CRITERIA}

Respond with only dictionary (same format as the given dictionary) with no extra or surrounding text.
Write [END] after you are done.

Dictionary with Paraphrased Sentences:

---

## Prompt for Generating an Instruction and Reference Answer

Your job is to generate a new novel problem and a response that is related to the given score rubric.

The score rubric:
{CRITERIA}

* Problem
- The problem should inherently be related to the score criteria and score rubric given above. Specifically, the score criteria should be the core attributes required to solve the problem.
- The problem itself should not be too generic or easy to solve.
- If the score rubric is related to logical abilities, generate problems that require math or coding abilities.
- Try to make the person who might solve the problem not notice the existence of the score rubric by not explicitly mentioning it, and also provide additional inputs and options if needed.
- Assume a situation where a user is interacting with an AI model. The user would try to ask in a first-person point of view, but not using terms like "I", "A User" or "You" in the first sentence.
- Do not give a role to the AI, assume that the user is asking a question from his point of view.
- Do not include any phrase related to AI model in the problem.

* Response
- The response should be a response that would get a score of 5 from the score rubric.
- The response should be as detailed as possible unless the score rubric is related to conciseness or brevity. It should consist of multiple paragraphs, a list of items, or a step-by-step reasoning process.
- The response should look like how a well-prompted GPT-4 would normally answer your problem.

* Format
- DO NOT WRITE ANY GREETING MESSAGES, just write the problem and response only.
- In front of the problem, append the phrase "Problem:" and in front of the response, append the phrase "Response:".
- Write in the order of "Problem" - "Response", where the two items are separated by the phrase "[NEXT]".
- Write [END] after you are done.

Data Generation:

## Prompt for Generating Responses and Feedback

Your job is to generate a response that would get a score of {SCORE} and corresponding feedback based on the given score rubric. For reference, a reference response that would get a score of 5 is also given.

Instruction:
{INSTRUCTION}

The score rubric:
{CRITERIA}

Reference response (Score 5):
{REFERENCE}

* Response
- The quality of the score {SCORE} response should be determined based on the score rubric, not by its length.
- The score {SCORE} response should have the same length as the reference response, composed of {SENT NUM} sentences.
- Do not explicitly state the keywords of the score rubric inside the response.

* Feedback
- The score {SCORE} feedback should each be an explanation of why the response would get a score of {SCORE}. It should be written based on the generated response and score rubric.
- The score {SCORE} feedback shouldn't just copy and paste the score rubric, but it should also give very detailed feedback on the content of the corresponding response.
- The score {SCORE} feedback should include the phrase "So the overall score is {SCORE}" in the last sentence.

* Format
- DO NOT WRITE ANY GREETING MESSAGES, just write the problem and response only.
- In front of the response, append the phrase "Response:" and in front of the feedback, append the phrase "Feedback:".
- Write in the order of "Response" - "Feedback", where the two items are separated by the phrase "[NEXT]".
- Write [END] after you are done.

Data Generation:

## K  PROMPT USED FOR PROMETHEUS

In this section, we provide the prompt used for training/inferencing PROMETHEUS. Note that after applying the prompt template shown below, we also apply Llama-2's basic conversation prompt template in order to minimize the discrepancy between the training process of Llama-2 and training on the FEEDBACK COLLECTION.

---

**Prompt for Prometheus**

###Task Description:
An instruction (might include an Input inside it), a response to evaluate, a reference answer that gets a score of 5, and a score rubric representing an evaluation criterion is given.
1. Write a detailed feedback that assesses the quality of the response strictly based on the given score rubric, not evaluating in general.
2. After writing a feedback, write a score that is an integer between 1 and 5. You should refer to the score rubric.
3. The output format should look as follows: "Feedback: (write a feedback for criteria) [RESULT] (an integer number between 1 and 5)"
4. Please do not generate any other opening, closing, and explanations.

###The instruction to evaluate:
{instruction}

###Response to evaluate:
{response}

###Reference Answer (Score 5):
{reference answer}

###Score Rubrics:
[{ criteria description }]
Score 1: {score1 description}
Score 2: {score2 description}
Score 3: {score3 description}
Score 4: {score4 description}
Score 5: {score5 description}

###Feedback:

---

## L  QUALITATIVE EXAMPLES OF GENERATED FEEDBACK

Figure 16, Figure 17, Figure 18, Figure 19 shows a qualitative example of feedback generated by either GPT-4, PROMETHEUS (13B), and Code-Llama trained on the FEEDBACK COLLECTION.

## M  EXPERIMENTAL DETAILS FOR HUMAN EVALUATION

The user interface used for human evaluation is shown in Figure 20. In order to acquire a score decision for the response, a decision of which feedback is better, and an annotation of why they made a decision to choose one of the feedback, we constructed the user interface in sequential order. Each annotator was informed not to solve the questions below before solving the ones above them. The test instances were randomly selected among FEEDBACK BENCH (Customized Unseen Rubric set), Vicuna Bench, and MT Bench except for math and coding instances. The experiments were conducted on LabelStudio, hosted on Huggingface Spaces. We asked each annotator to spend at least 3 hours annotating 45 instances each and paid $ 50. The 9 annotators were divided into 3 groups: (1) PROMETHEUS *vs* GPT-4, (2) PROMETHEUS *vs* GPT-3.5-Turbo, and (3) GPT-4 *vs* GPT-3.5-Turbo. The order of the feedback was randomly decided in order to prevent order bias during annotation.

**Score Rubric:**

**Does the model demonstrate an ability to logically process and interpret given information, apply relevant knowledge, and present an insightful conclusion?**

**Instruction**

Here is a dialogue between an assistant and an user.
Generate a human preferable response by the assistant.

😃 User: One morning after sunrise, Suresh was standing facing a pole. The shadow of the pole fell exactly to his right. Can you tell me the direction towards which the shadow was pointing - east, south, west, or north? Explain your reasoning steps.

🍚 Assistant:Based on the given information, Suresh was standing facing a pole, and the shadow of the pole fell exactly to his right. To determine the direction towards which the shadow was pointing, we need to consider the movement of the sun during the day. \n\nThe sun rises in the east and sets in the west. In the morning, when Suresh was standing facing the pole, the sun would have been in the eastern sky. Since the shadow of the pole fell exactly to Suresh's right, it means that the shadow was pointing towards the west.
Therefore, the direction towards which the shadow was pointing is west.

😃 User: To which direction was Suresh facing? How do you solve this?

🍚 Assistant:

**Response from ChatGPT**

Apologies for the confusion.
To determine the direction Suresh was facing, we need to consider the position of the shadow in relation to him. If the shadow fell exactly to his right, it means that Suresh was facing towards the west.
The shadow is formed when an object blocks the light from the sun. In this case, since the shadow is to Suresh's right, it indicates that the sun is in front of him, casting the shadow towards his back.
Therefore, Suresh was facing towards the east.

**GPT-4's Feedback**

The model interprets the information from the user accurately and applies the concept of the sun's position and its effect on the formation of shadows. However, it fails to reach the correct conclusion that Suresh is facing the south. Instead, it incorrectly concludes that Suresh is facing the east. The response demonstrates partial understanding of the problem but lacks complete logical coherence in the solution.

The final score is 3 out of 5.

**Prometheus's Feedback**

The response provided by the model is inaccurate and doesn't demonstrate a clear understanding of the given information and task. According to the user's query, Suresh was standing facing a pole, and the shadow of the pole fell exactly to his right. This implies that Suresh was facing towards the west, not the east as the model suggests. Furthermore, the model failed to properly explain the reasoning behind its conclusion, leading to an illogical conclusion that contradicts the initial information provided. The model didn't correctly apply the knowledge of the movement of the sun during the day, nor did it take into account the position of the shadow in relation to Suresh.

The final score is 1 out of 5.

Figure 16: An example of comparing the feedback generated by PROMETHEUS and GPT-4. GPT-4 was labeled to generate an abstract feedback.

**Score Rubric**
**Does the Response represent a balanced view of the factors?**

**Instruction**

Why might someone prefer to shop at a small, locally-owned business
instead of a large chain store, even if the prices are higher?

**Response from Vicuna 13B**

There are several reasons why someone might prefer to shop at a small, locally-owned business instead of a large chain store, even if the prices are higher:

1. Supporting the local economy: By shopping at a small, locally-owned business, you are supporting the local economy and helping to keep money within the community.
2. Personalized service: Small, locally-owned businesses often offer personalized service and attention that may not be available at a large chain store. The staff may be more knowledgeable about the products they sell and may be more willing to help you find what you're looking for.
3. Unique products: Small, locally-owned businesses often carry unique products that you may not be able to find at a large chain store. This can be especially true for businesses that specialize in niche products or that offer custom-made items.
4. Community involvement: Small, locally-owned businesses are often more involved in the local community than large chain stores. They may sponsor local events, support local charities, and be more connected to the community as a whole.
5. Better customer service: Small, locally-owned businesses often have a more personal relationship with their customers and may be more willing to go above and beyond to ensure customer satisfaction.
6. Environmental and ethical considerations: Some people prefer to shop at small, locally-owned businesses because they believe that these businesses are more likely to have environmentally and socially responsible practices.

Overall, shopping at a small, locally-owned business can be a way to support the local community, get personalized service, and find unique products that you may not be able to find at a large chain store.

**GPT-4's Feedback**

The response has done an excellent job of presenting a balanced view of the factors that might make someone prefer shopping at a small, locally-owned business instead of a large chain store, even if the prices are higher. Six different reasons were given, each of them explained thoroughly and in depth. No one reason was given more emphasis than the others, and each explanation was fair and unbiased. However, it's worth noting that the response failed to consider factors that might favor large chain stores, as the reference answer did.

The final score is 4 out of 5.

**Prometheus's Feedback**

The response does a good job of presenting the factors of why someone might prefer to shop at a small, locally-owned business, including supporting the local economy, personalized service, unique products, community involvement, better customer service, and environmental and ethical considerations. However, the response could improve in presenting all factors with equal rigor.

For example, the response does not delve deeply enough into the specifics of how shopping at a small, locally-owned business could support the local economy, nor does it explain how these businesses might be more involved in the community compared to large chain stores. The response could also further discuss how these factors might not be as important to all consumers, similar to how the reference response acknowledges the importance of factors like price, convenience, and product range in large chain stores.

The final score is 4 out of 5.

Figure 17: An example of comparing the feedback generated by PROMETHEUS and GPT-4. PROMETHEUS was labeled to generate an overly critical feedback.

**Score Rubric:**

**Can the model's response be understood by a non-expert in the subject?**

**Instruction**

Explain the process of photosynthesis in plants in a way that a middle school student could understand.

**Response from Simulated GPT-4**

Photosynthesis, essentially, is a biochemical reaction in which plants use sunlight (solar energy), water (H2O), and carbon dioxide (CO2) to produce glucose (C6H12O6) and oxygen (O2).

It's the key method that plants use for their nutritional requirements. To begin with, plants capture solar energy with the chlorophyll in their leaves.

Then, with the absorbed solar energy, the plant's chloroplasts take water from the ground and CO2 from the atmosphere to perform the complex photosynthesis process.

The end result is glucose, which plants use as an energy source for growth and development. Additionally, plants emit oxygen, which is beneficial for us as it's necessary for our respiration.

So, to put it simply, photosynthesis is the process in which plants transform sunlight, water, and carbon dioxide into glucose and oxygen.

**GPT-4's Feedback**

Your explanation of photosynthesis is decent and uses analogies to make some aspects more relatable to a non-expert, such as a middle school student. However, there are still areas that could be challenging for them. The use of the terms 'biochemical reaction', 'chlorophyll', 'chloroplasts', and the chemical formulas for water, carbon dioxide, and glucose could confuse a non-expert, making it harder for them to grasp the overall concept. Instead of diving straight into the complex aspects of photosynthesis, you might have started with a simpler and more relatable analogy. Additionally, providing a step-by-step description of the process would have made your explanation more digestible.

So the overall score is 3.

**Prometheus's Feedback**

The explanation given in this response is generally understandable, but there are several terms and concepts that might be difficult for a middle school student to grasp. For instance, terms like "chlorophyll," "chloroplasts", "photosynthesis", "solar energy", "glucose", and "oxygen" are used without sufficient contextual exp`lanations. While the main ideas of photosynthesis are conveyed, the language used could potentially be confusing for a non-expert. Additionally, the concept of plants using sunlight, water, and carbon dioxide to produce glucose and oxygen is described but lacks concrete, relatable analogies that would have helped simplify the explanation. The explanation could be improved by using simpler language and incorporating more accessible examples or metaphors to help illustrate the process.

So the overall score is 3.

Figure 18: An example of comparing the feedback generated by PROMETHEUS and GPT-4. Both evaluator LM generated a good feedback, and hence was labeled as a tie.

**Score Rubric**
**How well is the implementation in terms of its correctness, efficiency, documentation, and error handling?**

### Instruction

Develop a C++ program that reads a text file line by line and counts the number of occurrences of a specific word in the file.

### Response from Wizard LM 13B

Here is a C++ program that reads a text file line by line and counts the number of occurrences of a specific word in the file:

```
#include <iostream>
#include <fstream>
#include <string>

using namespace std;
int main() {
    // File name
    string filename = \"example.txt\";

    // If the file does not exist, create it
    if (!exists(filename)) {
        ofstream out(filename);
        out << \"No such file.\" << endl;
    }

    // Open the file in read mode
    ifstream in(filename, ios::in);
    // Count the number of occurrences of the word
    int count = 0;
    string word = \"your_word_here\";

    // Replace with the word you want to count
    while (in >> word) {
        if (word == word) {
            count++;
        }
    }
    // Print the result
    cout << \"The word \" << word << \" appears \" << count << \" times in the file.\" << endl;

    // Close the file
    in.close();

    return 0;
}
```

Here's how the program works:
1. The first few lines include the necessary header files and define the `main()` function.
2. The `filename` variable is defined to store the name of the file to be read.
3. The program checks if the file exists and creates it if it does not.
4. The file is opened in read mode using the `ifstream` class.
5. The program enters a loop that reads each line of the file.
6. Inside the loop, the program checks if the current line contains the word to be counted. If it does, the `count` variable is incremented.
7. After the loop exits, the program prints the result to the console.
8. Finally, the file is closed using the `close()` method of the `ifstream` object.

To use this program, simply replace the `filename` variable with the name of the file you want to read, and run the program. The output will be the number of times the specified word appears in the file.

**GPT-4's Feedback**

The implementation provided is not fully correct. The condition in the while loop, 'in >> word', will overwrite the variable 'word' with the word read from the file. This makes the following 'if' statement to always evaluate to true. Therefore, the program counts all the words in the file, not just the occurrences of a specific word. Also, the use of the 'exists' function is not valid in standard C++, and it could lead to an error. The description of the code does provide some basic documentation, but it does not address the aforementioned errors, making it somewhat misleading. Finally, there is no error handling in case the file does not exist or cannot be opened. Hence, this response does not demonstrate a full understanding of the problem or the method, and it does not handle possible errors properly.

The final score is 3 out of 5.

**Prometheus's Feedback**

The provided C++ program correctly implements a solution to read a text file line by line and count the number of occurrences of a specific word. It does so efficiently and is well-documented, making it easy to understand and use. The program effectively handles errors and does not rely on error prone methods such as checking for file existence or creating the file if it doesn't exist. The code is well-organized and well-structured, making it easy to follow and maintain. So the overall score is 5.

**Code-Llama (7B) Fine-tuned on the Feedback Collection**

Feedback:\n\nThe provided response meets some of the criteria in the score rubric, but there are areas for improvement. Here are some specific feedback points:

* Correctness: The response correctly implements the task of counting the number of occurrences of a specific word in a file. However, the code could benefit from additional error handling and handling of edge cases, such as dealing with files that do not exist or containing invalid data. (Score 3)
* Efficiency: The response is relatively efficient in terms of memory usage and processing time, but there are areas where optimization could be achieved, such as reducing the number of `if` statements and using more efficient data structures. (Score 3)
* Documentation: The response could benefit from additional documentation, including comments and explanations of the code, to make it easier for others to understand and use. (Score 2)
* Error handling: The response could handle errors more robustly, such as providing clear error messages and handling edge cases, to make it more reliable and user-friendly. (Score 3)

Overall, the response scores a 3 out of 5 in terms of the evaluation criteria.

Figure 19: An example of comparing the feedback generated by PROMETHEUS, GPT-4, and Code-Llama trained on the FEEDBACK COLLECTION. Compared to PROMETHEUS, using a base model specialized on the code domain also helps to criticize and evaluate responses within the code domain.

**Please read through a score rubric (채점 기준표), an instruction (AI 모델이 수행할 지시), a response (AI 모델의 답변), and a reference answer (모범 답안).**

Please refer to this score rubric during evaluation:

[Can the model's response be understood by a non-expert in the subject?]
Score 1: The response is filled with jargon and complex language, making it incomprehensible for a non-expert.
Score 2: The response includes some explanations, but still relies heavily on jargon and complex language.
Score 3: The response is somewhat clear, but could still be challenging for a non-expert to fully understand.
Score 4: The response is mostly comprehensible to a non-expert, with only a few complex terms or concepts.
Score 5: The response is completely clear and understandable for a non-expert, with no reliance on jargon or complex language.

🔴 The instruction to evaluate:

Explain the process of photosynthesis in plants in a way that a middle school student could understand.

🔵 Response to evaluate:

Photosynthesis, essentially, is a biochemical reaction in which plants use sunlight (solar energy), water ($H_2O$), and carbon dioxide ($CO_2$) to produce glucose ($C_6H_{12}O_6$) and oxygen ($O_2$). It's the key method that plants use for their nutritional requirements. To begin with, plants capture solar energy with the chlorophyll in their leaves. Then, with the absorbed solar energy, the plant's chloroplasts take water from the ground and $CO_2$ from the atmosphere to perform the complex photosynthesis process. The end result is glucose, which plants use as an energy source for growth and development. Additionally, plants emit oxygen, which is beneficial for us as it's necessary for our respiration. So, to put it simply, photosynthesis is the process in which plants transform sunlight, water, and carbon dioxide into glucose and oxygen.

🟢 Reference Answer (An answer that would get a score of 5):

Photosynthesis is like a food-making process for plants. Let's break it down and make it simple.

Imagine plants are like little chefs. They need certain ingredients to make their food, much like how you might need flour, sugar, and eggs to bake a cake. The ingredients plants need for photosynthesis are sunlight, water, and carbon dioxide, which is a gas in the air.

Here's how it works:

1. Plants use their leaves to capture sunlight. You can think of leaves as solar panels. The sunlight is the energy the plants need to start making their food.

2. Water is absorbed through the roots of the plant from the soil. Think of the roots as straws, sucking up the water from the ground.

3. Carbon dioxide comes from the air. Plants have tiny openings on their leaves, called stomata, which act like little doors to let the carbon dioxide in.

Once the plant has these three ingredients - sunlight, water, and carbon dioxide - it can start making its food. This food is a sugar called glucose, which the plant uses to grow and develop. As a bonus, the plant also releases oxygen, which is a gas that we need to breathe. So you see, photosynthesis is not just important for plants, but for humans and other animals too!

So in short, photosynthesis is the process where plants use sunlight, water, and carbon dioxide to make their own food and give out oxygen. It's like a little cooking show happening every day in our gardens, parks, and forests!

**What score would you give to the response above?**

- [ ] 1[1]
- [ ] 2[2]
- [ ] 3[3]
- [ ] 4[4]
- [ ] 5[5]

**Please read through FeedbackA (AI 모델의 답변에 대한 평가)**

Your explanation of photosynthesis is decent and uses analogies to make some aspects more relatable to a non-expert, such as a middle school student. However, there are still areas that could be challenging for them. The use of the terms 'biochemical reaction', 'chlorophyll', 'chloroplasts', and the chemical formulas for water, carbon dioxide, and glucose could confuse a non-expert, making it harder for them to grasp the overall concept. Instead of diving straight into the complex aspects of photosynthesis, you might have started with a simpler and more relatable analogy. Additionally, providing a step-by-step description of the process would have made your explanation more digestible. So the overall score is 3.

The final score is 3 out of 5.

**Please read through FeedbackB (AI 모델의 답변에 대한 평가)**

The explanation given in this response is generally understandable, but there are several terms and concepts that might be difficult for a middle school student to grasp. For instance, terms like "chlorophyll," "chloroplasts," "photosynthesis," "solar energy," "glucose," and "oxygen" are used without sufficient contextual explanations. While the main ideas of photosynthesis are conveyed, the language used could potentially be confusing for a non-expert. Additionally, the concept of plants using sunlight, water, and carbon dioxide to produce glucose and oxygen is described but lacks concrete, relatable analogies that would have helped simplify the explanation. The explanation could be improved by using simpler language and incorporating more accessible examples or metaphors to help illustrate the process. So the overall score is 3.

The final score is 3 out of 5.

**Which feedback is better?**

- [ ] feedbackA is better:+1:[6]
- [ ] feedbackB is better:raised_hands:[7]
- [ ] Tie:handshake:[8]
- [ ] Both are Bad:-1:[9]

**Why did you chose one of the feedback above? Choose more than one options.**

- [ ] The rejected feedback is unrelated to the given score rubric.[0]
- [ ] The rejected feedback is not relevant with the response.[q]
- [ ] The rejected feedback is too general and abstract.[w]
- [ ] The rejected feedback is overly critical.[e]
- [ ] The rejected feedback is overly optimistic.[t]
- [ ] The rejected feedback is not consistent with the given score it gave.[a]
- [ ] None of the above.[s]

Figure 20: The annotation user interface for labeling the human scores, pairwise comparison of the two feedback and gathering labels of why one feedback was preferred over the other one.

