# OpenReview forum: "Prometheus: Inducing Fine-Grained Evaluation Capability in Language Models"
_ICLR.cc/2024/Conference — ICLR 2024 poster_

### Official Review · Reviewer_WpqT · 2023-11-01

**Soundness:** 2 fair
**Presentation:** 3 good
**Contribution:** 3 good
**Rating:** 5
**Confidence:** 3

**Summary:**

This paper introduces PROMETHEUS, an open-source large language model (LLM) that aims to provide evaluation capabilities on par with the proprietary GPT-4. To achieve this, the authors create a new dataset called FEEDBACK COLLECTION, containing diverse and fine-grained user assessment criteria. PROMETHEUS is trained using this dataset and demonstrates a strong correlation with GPT-4's evaluation capabilities, as well as human evaluators.

This paper addresses the limitations of using proprietary LLMs like GPT-4 for evaluation, such as closed-source nature, uncontrolled versioning, and prohibitive costs. The PROMETHEUS aims to offer an alternative that is open-source, reproducible, and cost-effective. The FEEDBACK COLLECTION dataset allows the model to generalize to various evaluation preferences and real-world scenarios. In tests, PROMETHEUS outperforms other baselines and shows potential as a universal reward model.

**Strengths:**

1. The organization of this paper is well-structured, making it easy to read and comprehend.
2. This work presents the creation of a FEEDBACK COLLECTION dataset, which encompasses a diverse range of scoring criteria, reference answers, and feedback. Based on this, an evaluation LLM is trained for assessing the text generated by large language models.
3. The analysis in this work is thorough, discussing the selection of base models, data construction, and demonstrating the importance of reference answers. This provides valuable insights for the evaluation of large models in the field.

**Weaknesses:**

1. The main contribution of this paper is the FEEDBACK COLLECTION dataset. However, the dataset has not been made publicly available, and the construction details are unclear. For instance, the content of Step 2 is incomplete, and Step 3 is overly simplistic. Furthermore, the prompts used during construction have not been disclosed.
2. Assessing the consistency of scores alone is insufficient; it is also necessary to evaluate the feedback corresponding to these scores. On one hand, it is important to determine whether the feedback aligns with the scoring criteria. On the other hand, it should be examined if the feedback can be appropriately matched with the given scores. In fact, humans are not solely interested in obtaining a score; they are more concerned with the feedback associated with that score, which can further guide the large language model to generate desired answers.
3. It is unclear whether the test data in Table 1 is manually constructed or generated by GPT4-0613. If it is generated by GPT4-0613, why are the Pearson/Kendall/Spearman evaluation metrics not equal to 1?
4. For the Unseen FEEDBACK COLLECTION Testset, should all unseen instances be extracted? If the evaluation is conducted by combining the 50 Unseen samples with the 1000 samples similar to the training distribution, would this overshadow the true performance when facing unseen distribution during training?

**Questions:**

see above

---

> ### Author Response · Authors · 2023-11-11
> **Rebuttal for Dataset Publicity Issues, Construction Details of the Feedback Collection and Detailed Analysis of the Language Feedback**
>
> Dear reviewer WpqT,
>
> We appreciate your comments and review of the paper.
>
> (W1) - (W4) is our response to the “Weaknesses”. Please note that we have updated our draft of the paper, so please refer to the index of Tables and Figures in the updated version.
>
> ---
>
> ### **(W1) Dataset Publicity Issues and Construction Details of the Feedback Collection**
>
> #### [**Dataset Publicity Issues**]
> Regarding our dataset, we couldn’t upload our file due to its large size (OpenReview has a 100MB limit) and we didn’t include the link due to the anonymous policy (it is currently open-source at a public website). We will include the link in our camera-ready version. Also, we have uploaded our anonymized version of the code as supplementary material.
>
> #### [**Details for Dataset Construction Process**]
>
> Here is a detailed explanation of the dataset construction process in the updated draft (Section 3.1) :
>
> The collection process consists of (1) the curation of 50 initial seed rubrics, (2) the expansion of 1K new score rubrics through GPT-4, (3) the augmentation of realistic instructions, and (4) the augmentation of the remaining components in the training instances (i.e. responses including the reference answers, feedback, and scores). Figure 6 shows the overall augmentation process.
>
> * Step 1: Creation of the Seed Rubrics We begin with the creation of a foundational seed dataset
> of scoring rubrics. Each author curates a detailed and fine-grained scoring rubric that each personnel considers pivotal in evaluating outputs from LLMs. This results in an initial batch of 50 seed rubrics.
>
> * Step 2: Augmenting the Seed Rubrics with GPT-4 Using GPT-4 and our initial seed rubrics, we
> expand the score rubrics from the initial 50 to a more robust and diverse set of 1000 score rubrics. Specifically, by sampling 4 random score rubrics from the initial seed, we use them as demonstrations for in-context learning (ICL), and prompt GPT-4 to brainstorm a new novel score rubric. Also, we prompt GPT-4 to paraphrase the newly generated rubrics in order to ensure PROMETHEUS could generalize to the similar score rubric that uses different words. We iterate the brainstorming → paraphrasing process for 10 rounds. The detailed prompt used for this procedure is in Appendix J.
>
> * Step 3: Crafting Novel Instructions related to the Score Rubrics With a comprehensive dataset of 1000 rubrics at our disposal, the subsequent challenge was to craft pertinent training instances. For example, a score rubric asking “Is it formal enough to send to my boss” is not related to a math problem. Considering the need for a set of instructions closely related to the score rubrics, we prompt GPT-4 to generate 20K unique instructions that are highly relevant to the given score rubric.
>
> * Step 4: Crafting Training Instances Lastly, we sequentially generate a response to evaluate and corresponding feedback by prompting GPT-4 to generate each component that will get a score of i (1 ≤ i ≤ 5). This leads to 20 instructions for each score rubric, and 5 responses & feedback for each instruction. To eliminate the effect of decision bias when fine-tuning our evaluator LM, we generate an equal number of 20K responses for each score. Note that for the response with a score of 5, we generated two distinctive responses so we could use one of them as an input (reference answer).
>
> For the prompts used for each step, please refer to Appendix J in the updated draft. We hope the updated version will waive your concern regarding the lack of details.
>
> ---
>
> ### **(W2) Detailed Analysis of the Feedback**
>
> We strongly agree that language feedback often conveys richer information compared to just looking at the scoring decision for evaluation. We have included the results regarding the examination of the feedback quality from Prometheus and GPT-4, which is shown in Figure 5.
>
> When human annotators assessed the failure cases of Prometheus and GPT-4, `the **semantic inconsistencies** between the **“score decision & feedback”** (GPT-4: 2.00%, Prometheus: 2.86%) or the **“score rubric & feedback”** (GPT-4: 8.00%, Prometheus: 5.71%) wasn’t the main reason.
>
> We are thrilled towards the direction of extracting meaningful insights from the language feedback which could be explored in future work. In this aspect, we found that existing open-source models such as Llama-2-Chat (70B) often generate abstract feedback that is meaningless and GPT-4 might be too expensive to use as it took $8,000 to construct the Feedback Collection.
>
> We hope Prometheus could be used as a reliable and inexpensive source for further research in this direction.

---

> > ### Author Response · Authors · 2023-11-11
> > **Rebuttal for GPT-4's Discrepancy and Explanation of the Feedback Collection Test Set**
> >
> > ---
> >
> > ### **(W3) GPT-4’s discrepancy between Generating the Data / Solving it again**
> > The test dataset of the Feedback Collection was constructed using GPT-4-0613. The detailed process is identical to how the training set was constructed. Yet, the difference is that we ensured that the scoring criteria does not overlap with the training set as shown in Figure 7 (Left skewed Rouge Score distribution).
> >
> > We conjecture that the reason why GPT-4-0613 doesn’t acquire a correlation of 1.0 is that there is a **discrepancy between generating a new instruction & evaluating the problem it generated**. On the other hand, Prometheus was specifically trained on such instruction, hence we conjecture that the **distribution is shifted narrowly towards the Feedback Collection** that GPT-4-0613 generated.
> >
> > We have included this information in Appendix C.3.
> >
> > ---
> >
> > ### **(W4) Explanation of the Feedback Collection Test Set**
> > We have not fully understood the point of this question, hence we would kindly like to request whether you could elaborate on it.
> >
> > For reference, the following is a more detailed explanation of how the seen and unseen subset within the test set differs.
> >
> > #### **[Difference Between the Seen (Rubric) Subset and the Unseen (Rubric) Subset**
> >
> > * Both the seen and unseen subset contains unseen instructions and responses compared to the training set. The difference is whether the score rubric was seen during training.
> >
> > * Seen (Rubric) Test set: This subset contains the exact 1K score rubrics within the training data. Yet, it has 1K novel instructions and responses that are not included in the training data.
> >
> > * Unseen (Rubric) Test set: This subset contains 50 novel score rubrics that are not contained within the training data. It also contains 1K instructions and responses that are not included in the training data.
> >
> > * Since both the Seen (Rubric) and Unseen (Rubric) have **non-overlapping instructions and responses** compared to the training data, we argue that this is not problematic during the evaluation process of Prometheus.
> >
> > We have included this information in Appendix B.1 in the updated draft.

---

> > > ### Author Response · Authors · 2023-11-20
> > > **Have you read our response?**
> > >
> > > Dear Reviewer WpqT,
> > >
> > > We understand the discussion timeline is inconvenient for your busy schedule, but we would love to continue our discussion before the time window closes.
> > >
> > > We hope that we were able to resolve all your questions and please let us know if there's more.

---

> > > > ### Author Response · Authors · 2023-11-23
> > > > **Appreciation for your review**
> > > >
> > > > Dear Reviewer WpqT,
> > > >
> > > > Thank you for recognizing the strengths of our work, including the versatility and academic value of Prometheus, and its comparative effectiveness against GPT-4.
> > > >
> > > > We believe that we were able to resolve your concerns on the following aspects:
> > > > - Dataset Publicity Issues and Construction Details of the Feedback Collection
> > > > - Detailed Analysis of the Feedback
> > > > - GPT-4’s discrepancy between Generating the Data / Solving it again
> > > > - Explanation of the Feedback Collection Test Set
> > > >
> > > > Given these substantial improvements, we respectfully ask if you could reconsider the score, in light of these revisions. Your insights have been invaluable in making our paper better, and we hope our updates meet the conference's high standards.
> > > >
> > > > Thank you again for your time and valuable feedback.

---

### Official Review · Reviewer_LTFo · 2023-11-01

**Soundness:** 4 excellent
**Presentation:** 4 excellent
**Contribution:** 3 good
**Rating:** 6
**Confidence:** 4

**Summary:**

This paper presents Prometheus, an open-source language model that provides fine-grained evaluation capabilities comparable to GPT-4. The authors aim to overcome the challenges of using GPT-4 as an evaluator, such as its closed-source nature, uncontrolled versioning, and high cost. Prometheus is trained on a new dataset, the Feedback Collection, which includes a wide range of user-based evaluation criteria. The model shows strong correlation with GPT-4 evaluation on seven benchmarks and outperforms ChatGPT in human evaluation. Remarkably, Prometheus demonstrates a win rate of 58.62% when compared to GPT-4 evaluation and a win rate of 79.57% when compared to ChatGPT evaluation.

**Strengths:**

1. Prometheus can assess responses based on novel and unseen score rubrics and reference materials provided by the user. This flexibility makes it applicable to a variety of real-world criteria.
2. Prometheus can be freely used and further enhanced by the academic community, facilitating transparency and reproducibility.
3. Prometheus shows remarkable performance in comparison with GPT-4 in terms of evaluation capabilities and the quality of generated feedback.
4. The creation of the Feedback Collection, a dataset designed specifically for the task of teaching fine-grained evaluation to language models, is a significant contribution.

**Weaknesses:**

1. One of my concerns about this work is whether can Prometheus be generalized to other fields since the downstream benchmarks are close the the training data. More results on unseen data and more specific domains can better improve this work.

2. Potential bias of Prometheus. Can Prometheus be attacked by some adversarial attack methods? Does it have stronger biases like length bias compared with GPT-4?

3. Dependency on GPT-4 Feedback: The training of Prometheus relies heavily on feedback generated by GPT-4. The model's ability to generalize beyond the feedback patterns of GPT-4 is unclear.

**Questions:**

See weaknesses.

---

> ### Author Response · Authors · 2023-11-11
> **Rebuttal for Generalization to Other Fields**
>
> Dear reviewer LTFo,
>
> We appreciate your comments and review for the paper.
>
> (W1) - (W3) is our response to the “Weaknesses”. Please note that we have made significant updates to the organization of the paper (content is same), so please refer to the index of Tables and Figures from the updated version.
>
> ---
>
> ### **(W1) Generalization to Other Fields**
>
> #### [**Difference Between the Feedback Collection and the Datasets used for Evaluation**]
> We would first like to emphasize that the datasets used for experiments in Table 3 (Vicuna Bench, MT Bench, Flask) contains instructions that have different characteristics with the training dataset we used (Feedback Collection).
>
> Specifically, the dataset in Table 4 have the following characteristics:
>
> * **Vicuna Bench**:Relatively short instructions that are mostly consisted of How/What type of questions in addition to some coding and math instructions. (e.g., "How can governments utilize fiscal and monetary policies to combat economic recessions?)
>
> * **MT Bench**: The 2nd instruction we used in this multi-turn dataset mostly asks for revision or further explanation of the previous response. (e.g., “Now, do the same task again but only use four-word sentences.”)
>
> * **Flask Eval**: Contains academic NLP tasks such as classification datasets (e.g., MMLU, FEVER) or the train split of instruction datasets (e.g., Self-Instruct, WizardLM) (e.g., "Paraphrase the given text in an academic style. Input: Lots of papers have been published on this topic.”)
>
> In contrast, the **Feedback Collection** was constructed with the main consideration of a very detailed, realistic situation where a user is interacting with an AI. (e.g., “Suppose there is a friend who is feeling low due to a poor performance on a recent exam. The friend is now seeking advice, encouragement, and a bit of humor to lighten the mood. How would one approach this situation, incorporating wit and a playful tone to uplift the friend's spirit?”)
>
> The point we would like to emphasize is that Prometheus shows a higher correlation with both human evaluators (Figure 3) and GPT-4 (Table 3) in all of these different datasets. This supports that Prometheus could generalize to other instruction evaluation settings.
>
> #### [**Prometheus also works in Coarse-grained Relative Evaluation Setting although it was trained for Fine-grained Aboslute Evaluation**]
>
> Also, we would like to emphasize that the human preference dataset experiments in Table 4 (HHH Alignment, MT Bench Human Judgment) is evidence indicating the generalizability of Prometheus.
>
> Specifically, we would like to emphasize the following two points:
>
> * Results show that Prometheus could **generalize to coarse-grained criteria** (Helpfulness, Harmlessness, Honesty) although it was trained with detailed and fine-grained criteria (Cultural Sensitivity, Humorous, Considering regulation and compliance requirements).
>
> * Results show that while Prometheus was trained in an absolute scoring setting, its evaluation capabilities could also be **transferred to a relative scoring setting**. Prometheus outperforms open-source reward models that were specifically trained with human preference datasets (StanfordNLP Reward Model, ALMOST) and even GPT-3.5-Turbo.  Although it doesn’t get access to both responses, it could manage to make a score decision that gives a higher score to human-preferred responses. Without any generalization, this would be extremely hard.
>
> ---

---

> ### Author Response · Authors · 2023-11-11
> **Rebuttal for Potential Bias of Prometheus and Dependency on GPT-4 Feedback**
>
> ---
>
> ### **(W2) Potential bias of Prometheus**
>
> #### [**Absolute Scoring is less vulnerable to Length Bias**]
> We strongly agree that your concern regarding various biases that could occur during evaluation is very crucial in the field. As you specifically mentioned about **length bias**, we would like to discuss more about it in detail.
>
> We would like to highlight that in this aspect, evaluation based on **Absolute Scoring holds a strong advantage over Relative Scoring** since the evaluator LM doesn’t get vulnerable to the length differences or other surface patterns between the two responses during evaluation.
>
> In Figures 13,14 and 15, we have measured whether Prometheus or GPT-4 gave a higher score to longer responses in an absolute evaluation setting and found that it is not the case. The responses that got a score of 1 - 5 all have similar score distributions, supporting the idea that Absolute Scoring might be a good way to mitigate length biases.
>
> #### [**Additional Experiment on Adversarial Dataset**]
>
> To further analyze other datasets, we have conducted experiments with the recently proposed LLMBar dataset [1]. This dataset has an adversarial subset where the better response is shorter and the worse response is longer. With the same setting as in our human preference datasets in Table 4, we have conducted additional experiments and obtained the following results:
>
> |          | Random Guess | Llama-2-Chat (70B) | GPT-3.5-Turbo-0613 | Prometheus-13B | GPT-4-0613 |
> |----------|--------------|--------------------|--------------------|----------------|------------|
> | Accuracy | 48.91        | 35.87              | 44.57              | 63.04          | 75.00      |
>
> This result supports that Prometheus is relatively robust to adversarial patterns such as length bias compared to other baselines such as Llama-2-Chat (70B) and GPT-3.5-Turbo-0613.
>
> ---
>
> ### **(W3) Dependency on GPT-4 feedback**
>
> #### [**Justification on the reliance towards GPT-4**]
>
> We strongly agree that the concern you have raised is a very important point not only for our work but also for the overall community in general since a lot of recent work heavily relies on data augmented from GPT-4.
>
> Nonetheless, augmentation based on GPT-4 has the following advantages relevant to the strengths you have mentioned:
>
> * GPT-4 shows remarkable performance in terms of Evaluation, and previous work [2,3,4] has also highlighted that GPT-4 could closely emulate human evaluators.
>
> * For performing fine-grained evaluation with detailed criteria, the only source we could rely on was either GPT-4 or human evaluators since other LLMs could not flexibly ground on the given score rubric. While we approximately spent $8,000 to create the Feedback Collection, collecting the same amount with only humans will cost at least 10x, especially because the level of expertise required for such annotations is very high.
>
> * While our construction process heavily relies on GPT-4, it also opens the door for other researchers to build upon our work and catch up since they could fully utilize the open-source dataset and model instead of paying another $8,000 through OpenAI API access.
>
> On the other hand, we think that exploring different strategies such as mixing with human feedback or other types of model feedback (when it exists) is a worthwhile future work to explore.
>
> #### [**Difference of the Language Feedback Pattern between Prometheus and GPT-4**]
>
> Lastly, we would like to highlight that although Prometheus was trained on data from GPT-4, it shows quite a different feedback pattern compared to that of GPT-4.
>
> In Figure 5, when inspecting the quality of the feedback, we observed that human annotators determined that GPT-4 generated relatively abstract and generic feedback, while Prometheus generated either too critical or too optimistic feedback.
>
> Hence, even though the training data was generated by GPT-4, the results show that they can behave differently. This may not be a conclusive answer to your concern, but we hope it clears some of it.
>
> ---
>
> ### **References**
>
> [1] Zeng, Z., Yu, J., Gao, T., Meng, Y., Goyal, T. and Chen, D., 2023. Evaluating large language models at evaluating instruction following. arXiv preprint arXiv:2310.07641.
>
> [2] Zheng, L., Chiang, W.L., Sheng, Y., Zhuang, S., Wu, Z., Zhuang, Y., Lin, Z., Li, Z., Li, D., Xing, E. and Zhang, H., 2023. Judging LLM-as-a-judge with MT-Bench and Chatbot Arena. arXiv preprint arXiv:2306.05685.
>
> [3] Dubois, Y., Li, X., Taori, R., Zhang, T., Gulrajani, I., Ba, J., Guestrin, C., Liang, P. and Hashimoto, T.B., 2023. Alpacafarm: A simulation framework for methods that learn from human feedback. arXiv preprint arXiv:2305.14387.
>
> [4] Ye, S., Kim, D., Kim, S., Hwang, H., Kim, S., Jo, Y., Thorne, J., Kim, J. and Seo, M., 2023. Flask: Fine-grained language model evaluation based on alignment skill sets. arXiv preprint arXiv:2307.10928.

---

> > ### Author Response · Authors · 2023-11-20
> > **Have you read our response?**
> >
> > Dear Reviewer LTFo,
> >
> > We understand the discussion timeline is inconvenient for your busy schedule, but we would love to continue our discussion before the time window closes.
> >
> > We hope that we were able to resolve all your questions and please let us know if there's more.

---

> > > ### Comment · Reviewer_LTFo · 2023-11-22
> > > **Thank you for your response**
> > >
> > > Dear Authors,
> > >
> > > Thank you for your response and responding to the question.
> > >
> > > Good luck !!

---

> > > > ### Author Response · Authors · 2023-11-22
> > > > **Appreciation for Repliance**
> > > >
> > > > Dear reviewer LTFo,
> > > >
> > > > Thank you for recognizing the strengths of our work, including the versatility and academic value of Prometheus, and its comparative effectiveness against GPT-4.
> > > >
> > > > We have diligently addressed your concerns on generalization, length bias analysis, and Prometheus's unique capabilities, even when trained on data from GPT-4. These enhancements directly align with the strengths you've appreciated.
> > > >
> > > > Given these substantial improvements, we respectfully request you to reconsider the score, in light of these revisions. Your insights have been invaluable, and we hope our updates meet the conference's high standards.
> > > >
> > > > Thank you again for your time and valuable feedback.

---

### Official Review · Reviewer_B7Vr · 2023-11-02

**Soundness:** 1 poor
**Presentation:** 1 poor
**Contribution:** 1 poor
**Rating:** 1
**Confidence:** 4

**Summary:**

Paper presents a new benchmark for building evaluation systems with LLMs. Although the paper contribution is promising, there are some serious problems in the paper. Many of the figures are missing and unvisible. The paper contribution, whether this is a novel LLM, or a data set generated by gpt-4 is unclear. The model is advertised as open-source but how the data will be shared is unstated. If an LLM is built on this data, which is described as a 100K synthesized data set, how is it an 13B LM is unclear. Paper cannot be published in such state with so much missing information.

**Strengths:**

Proposes open-source LLM for evaluation

**Weaknesses:**

Model implementation is not described.
Experimental methodology not clear or supported.
Most figures missing.
Contribution too small (not any new data, model or any advertised contribution is clearly described).
Data is synthetic and not corrected by humans for any potential errors.

**Questions:**

Where is Figure 2?
Where is Figure 4?

---

> ### Author Response · Authors · 2023-11-11
> **Issues with the Figures**
>
> Dear reviewer B7Vr,
> We appreciate the review and comments on the paper.
>
> Before addressing the other issues, I think the problem of the missing figure might be due to your browser.
> I had the same problem before when using Safari. The draft size is quite large and Safari doesn't seem to render it very well.
>
> Could you try another browser or environment to check if the problem holds?
> Currently, Figure 2 and Figure 4 are included in the draft.

---

> ### Author Response · Authors · 2023-11-11
> **Clarification of the Implementation, Experimental Methodology and Contribution**
>
> Dear reviewer B7Vr,
>
> We appreciate your comments and review for the paper.
>
> (W1) - (W4) are our responses to the weaknesses you have mentioned. Please note that we have made significant updates to the organization of the paper (content is same), so please refer to the index of Tables and Figures from the updated version.
>
> ---
>
> ### **(W1) Details of Model Implementation**
>
> Our explanation of the model implementation is included in Section 3.2 and Appendix F. Using the llama-recipes repository, we have fine-tuned Llama-2-Chat (7B & 13B) to obtain Prometheus using 8 A100 (80GB) GPUs with FSDP. The training is similar to Chain-of-Thought Fine-tuning, where we fine-tune to sequentially generate the feedback and the the score.
>
> The hyperparameters we used during training and inference are included in Table 8 and 9.
>
> ---
>
> ### **(W2) Explanation of the Experimental Methodology**
>
> Since our main objective is to obtain a open-source evaluator language model that could closely emulate GPT-4 and human evaluators, we mainly divide our experiments into two different phases. Further details are included in Appendix B.
>
> * We first check how the score is similar with either human evaluators (as shown in Figure 3) and GPT-4 (as shown in Table 2 and 3). Specifically, after parsing the score decision, we compare with Pearson, Spearman, and Kendall-Tau correlation with the opponent.
> * Next we check how the other component that Prometheus generated, which is the language feedback has good quality. This is shown in Figure 4 along with an analysis in Figure 5. Specifically, we ask human evaluators to compare the language feedback and choose among Prometheus and GPT-4.
>
> ---
>
> ### **(W3) Explanation of our Contribution**
>
> The main contributions of our work are as follows:
>
> * We introduce the FEEDBACK COLLECTION dataset specifically designed to train an evaluator LM. Compared to previous feedback datasets, it includes customized scoring rubrics and reference answers in addition to the instructions, responses, and feedback.
> * We train PROMETHEUS, the first open-source LLM specialized for fine-grained evaluation that can generalize to diverse, real-world scoring rubrics beyond a single-dimensional preference such as helpfulness and harmlessness.
> * We conduct extensive experiments showing that by appending reference materials (reference answers, fine-grained score rubrics) and fine-tuning on feedback, we can induce evaluation capability into language models. PROMETHEUS shows high correlation with human evaluation, GPT-4 evaluation in absolute scoring settings, and also shows high accuracy in
> ranking scoring settings.
>
> ---

---

> > ### Author Response · Authors · 2023-11-13
> > **Explanation for the Synthetic Nature of the Feedback Collection**
> >
> > ---
> >
> > ### **(W4) Explanation for the Synthetic Nature of the Data**
> >
> > #### [**Prometheus shows high correlation with human evaluators**]
> >
> > Your concern regarding the synthetic nature of the Feedback Collection is valid. This was also one of our main concerns, which is why we conducted correlation with human evaluators in the MT Bench, VicunaBench, and Feedback Collection test set (shown in Figure 3). In addition to the score decision, human evaluators considered the feedback generated by Prometheus to be as valid as GPT-4 (shown in Figure 4).
> >
> > In addition, we have conducted experiments with 200 instances from the Flask Eval dataset to measure the correlation with human evaluators (an extension from Table 3).
> >
> > | Models             | Pearson Correlation w/ Human Evaluators | Pearson Correlation w/ GPT-4-0613 |
> > |--------------------|-----------------------------------------|-----------------------------------|
> > | Llama-2-Chat-7B    | 0.298                                   | 0.271                             |
> > | Llama-2-Chat-13B   | 0.275                                   | 0.265                             |
> > | Llama-2-Chat-70B   | 0.317                                   | 0.336                             |
> > | Prometheus-7B      | 0.327                                   | 0.367                             |
> > | Prometheus-13B     | **0.451**                               | **0.467**                         |
> > | GPT-3.5-Turbo-0613 | 0.421                                   | 0.422                             |
> > | GPT-4-0613         | 0.729                                   | 0.835 (Self-Consistency)          |
> >
> > This results support that although Prometheus was trained on a Synthetic dataset (Feedback Collection), it shows a higher correlation with human evaluators compared to other baselines such as GPT-3.5-Turbo and Llama-2-Chat (70B).
> >
> > #### [**Prohibitive Costs**]
> >
> > We would also like to emphasize that although manually inspecting all of the instances might be the best way to correct potential errors, this would be extremely expensive to do. We approximately spent $8,000 to create the Feedback Collection, collecting or revising the same amount with only humans will cost at least 10x, especially because the level of expertise required for such annotations is very high.
> >
> > #### [**Related Works also use GPT-4 to augment training data**]
> >
> > Lastly, we would like to note that a lot of recent work [1,2,3,4] heavily relies on datasets augmented from Proprietary LLMs as it is cost-efficient and as effective as obtaining good quality data compared to human annotation.
> >
> > ---
> >
> > ### References
> > [1] Wang, Y., Kordi, Y., Mishra, S., Liu, A., Smith, N.A., Khashabi, D. and Hajishirzi, H., 2022. Self-instruct: Aligning language model with self generated instructions. arXiv preprint arXiv:2212.10560.
> >
> > [2] Peng, B., Li, C., He, P., Galley, M. and Gao, J., 2023. Instruction tuning with gpt-4. arXiv preprint arXiv:2304.03277.
> >
> > [3] Taori, R., Gulrajani, I., Zhang, T., Dubois, Y., Li, X., Guestrin, C., Liang, P. and Hashimoto, T.B., 2023. Stanford alpaca: An instruction-following llama model.
> >
> > [4] Schick, T., Dwivedi-Yu, J., Dessì, R., Raileanu, R., Lomeli, M., Zettlemoyer, L., Cancedda, N. and Scialom, T., 2023. Toolformer: Language models can teach themselves to use tools. arXiv preprint arXiv:2302.04761.
> >
> > ---
> >
> > We hope we can have a fruitful discussion in detail after the figure rendering issue is resolved.

---

> > > ### Author Response · Authors · 2023-11-20
> > > **Have you read our response?**
> > >
> > > Dear Reviewer B7Vr,
> > >
> > > We understand the timeline is not convenient for your busy schedule, but we would appreciate if you could re-read our paper using Chrome. We know this is a bit inconvenient for you, but we weren't able to find a solution of the paper not rendering in Safari environment.
> > >
> > > We hope that you were able to resolve the figure rendering issue.

---

> > > > ### Author Response · Authors · 2023-11-23
> > > > **Figure rendering issue has been resolved.**
> > > >
> > > > Dear Reviewer B7Vr,
> > > >
> > > > We were able to finally resolve the figure rendering issue, and we have uploaded the revision. It was due to an error from Figma from converting SVG to PDFs.
> > > >
> > > > We thank you for your patience and hope that you could revise the scores accordingly after reading our response and the revised version.

---

### Official Review · Reviewer_6KgK · 2023-12-06

**Soundness:** 2 fair
**Presentation:** 3 good
**Contribution:** 3 good
**Rating:** 6
**Confidence:** 4

**Summary:**

This paper propose a method to automatically generate evaluation dataset using the few-shot capabilities of GPT-4 starting from "50 initial seed rubrics". The method shares some similarity to self-instruct (Wang et al., 2023). The main difference is apart from expanding seed rubrics, the proposed method also need to craft instructions and training instances. The resulting evaluation dataset (i.e., FEEDBACK COLLECTION) is further used to fine-tune a llama2-chat model, which is called PROMETHEUS. Essentially, PROMETHEUS tries to distill the evaluation capabilities of GPT-4.

Experiments show PROMETHEUS correlates well with human judgements and GPT-4 references.

**Strengths:**

- LLM based evaluation is important, since long text generation is becoming more difficult for human evaluators
- Using a distilled evaluator can significantly reduce cost, compared to GPT-4
- Results look good

**Weaknesses:**

- It is unclear why PROMETHEUS correlates better with human evaluation than GPT-4 on MT Bench (Figure 3), given the fact that PROMETHEUS distills from GPT-4. Further analysis is required.
- There is no details on how the seed examples are constructed.
- Generalization to different domains seems difficult. Note that this is not a weakness, since almost all LLMs have this problem. Let us assume that the GPT-4 evaluator is good enough (that is also the reason why the proposed method intend to distill from GPT-4). Now the task becomes how good can PROMETHEUS mimic GPT-4 evaluation.  It is shown in Tables 2 and 3 that the correlation between PROMETHEUS and GPT-4 reference is significantly higher in the in-domain dataset (data generated by GPT-4, as shown in Table 2) compared to out-of-domain datasets (refer to Table 3), with approximate values of 0.46 versus 0.86. Nevertheless, the development of an LLM evaluator that performs effectively on certain tasks is still meaningful.

**Questions:**

- To what extent do the seed examples (and their generated instances) resemble examples in VICUNA, MT Bench, or FLASK EVAL?

---

### Author Response · Authors · 2023-11-22
**General Response to All Reviewers**

We sincerely appreciate the significant effort and time committed by every reviewer in offering insightful and constructive questions regarding the paper. We want to provide a general response regarding the clarifications of the main contributions and how we addressed the common concerns.

### [**Main Contributions of the Paper**]
* We introduce the **Feedback Collection dataset** specifically designed to train an evaluator LM. Compared to previous feedback datasets, it includes customized scoring rubrics and reference answers in addition to the instructions, responses, and feedback.
* We train **Prometheus**, the first open-source LLM specialized for fine-grained evaluation** that can generalize to diverse, real-world scoring rubrics beyond a single-dimensional preference such as helpfulness and harmlessness.
* We conduct extensive experiments showing that by appending reference materials (reference answers, fine-grained score rubrics) and fine-tuning on feedback, we **show that we can induce evaluation capability into language models**. Prometheus shows a high correlation with human evaluation, GPT-4 evaluation in absolute scoring settings, and also shows high accuracy in ranking scoring settings.

### [**Updates in the Revised Draft**]
* **Technical Issue Resolution**: We acknowledge the issue highlighted regarding the rendering of the Figures in the Safari browser. We understand the importance of ensuring our paper's accessibility and readability across various platforms. In the current version of the draft, we have rectified this issue, ensuring that all figures are properly rendered and accessible in the Safari browser, in addition to other major web browsers.

* **Publicity of the Data, Model, and Code**:Addressing concerns from Reviewers B7Vr and WpqT about accessibility, we confirm the public availability of our dataset and model. The replication code is included in the supplementary material. Post-anonymity, we will integrate the relevant links.

* **Explanation of the Dataset Construction, Model Implementation, and Experimental Setting**: To alleviate concerns about our methodologies, raised by Reviewers B7Vr and WpqT, the revised draft delves into the creation of Feedback Collection (Section 3.1), details of Prometheus' training and evaluation (Section 3.2, Appendix C), and experimental setups (Sections 4.1 and 4.2).

* **Generalization to other Domains**: Addressing the universal concern about generalizability due to the synthetic nature of Feedback Collection, we restructured our experiments to assess Prometheus' alignment with human evaluators (Section 5.1), correlation with GPT-4 (Section 5.2), and the meaningfulness of its language feedback (Section 5.1). Using 3 datasets not included in training and two distinct ranking datasets, Prometheus demonstrated superior generalization and feedback quality compared to alternatives like ChatGPT and Llama-2-Chat (70B).

### [**Final Authors' Note**]
In our revised draft, we prioritized clarity and addressed all the highlighted concerns. We concur with the reviewers that our research offers significant insights into LLM evaluation and introduces a **novel methodology for assessing long-form LLM responses against diverse real-world standards**, such as cultural sensitivity and humor.

We emphasize that our work is a pioneering effort to **embed detailed evaluation capabilities into smaller models**. This approach addresses critical challenges associated with exclusive reliance on GPT-4, such as its closed-source nature, uncontrolled versioning, and high costs. Our work lays the groundwork for future research in LLM evaluation, advocating for user-defined criteria over traditional, coarse-grained benchmarks.

Thank you for your invaluable feedback and consideration.

---

### Meta-Review · Area_Chair_Zzio · 2023-12-06

**Metareview:**

This paper presents Prometheus, a language model evaluator that demonstrates notable strengths in assessing responses based on various criteria and reference materials, providing a significant contribution to the field of LLM evaluation. The creation of the Feedback Collection dataset is a valuable asset for fine-tuning and enhancing language models, and Prometheus's ability to generate meaningful feedback represents a promising direction in LLM development. However, the paper has several shortcomings that limit its suitability for an oral presentation but make it a good candidate for a poster session. The generalizability of Prometheus to different domains and fields remains unclear, particularly given its dependence on GPT-4 feedback for training and the potential biases it may inherit. Additionally, the lack of public availability of the Feedback Collection dataset and insufficient details on its construction. Despite these drawbacks, the paper is well-organized, and the thorough analysis provided offers valuable insights for the evaluation of large models.

**Justification For Why Not Higher Score:**

The paper is not accepted for an oral presentation primarily due to its limitations in generalizability and dependence on GPT-4 for training. The potential biases of Prometheus and its performance in unseen domains or specific fields are not adequately addressed, raising questions about its broader applicability. Furthermore, the lack of public availability of the Feedback Collection dataset and incomplete details on its construction detract from the paper's overall impact.

**Justification For Why Not Lower Score:**

This work presents significant contributions, including the novel Prometheus model and the creation of the Feedback Collection dataset. The model's performance in evaluation capabilities and the quality of generated feedback is noteworthy, and the paper's clear organization and comprehensive analysis provide valuable insights for the field. While the paper has shortcomings, its strengths in advancing the evaluation of large language models and the potential for further development and research in this area warrant its acceptance as a poster.

---

### Decision · Program_Chairs · 2024-01-16

Accept (poster)